# An isoform quantitative trait locus in SBNO2 links genetic susceptibility to Crohn's disease with defective antimicrobial activity

Dominik Aschenbrenner [1,13] ✉, Isar Nassiri [2,3,4,15], Suresh Venkateswaran[5,15], Sumeet Pandey[1,14], Matthew Page[6], Lauren Drowley[7], Martin Armstrong[8], Subra Kugathasan [5,15], Benjamin Fairfax [9,10,11,15] & Holm H. Uhlig [1,11,12] ✉

Despite major advances in linking single genetic variants to single causal genes, the significance of genetic variation on transcript-level regulation of expression, transcript-specific functions, and relevance to human disease has been poorly investigated. Strawberry notch homolog 2 (*SBNO2*) is a candidate gene in a susceptibility locus with different variants associated with Crohn's disease and bone mineral density. The SBNO2 locus is also differentially methylated in Crohn's disease but the functional mechanisms are unknown. Here we show that the isoforms of *SBNO2* are differentially regulated by lipopolysaccharide and IL-10. We identify Crohn's disease associated isoform quantitative trait loci that negatively regulate the expression of the non-canonical isoform 2 corresponding with the methylation signals at the isoform 2 promoter in IBD and CD. The two isoforms of SBNO2 drive differential gene networks with isoform 2 dominantly impacting antimicrobial activity in macrophages. Our data highlight the role of isoform quantitative trait loci to understand disease susceptibility and resolve underlying mechanisms of disease.

Genome-wide association studies (GWAS) have identified over 133,441 associations but only a minor fraction of these have been resolved to a single variant and even fewer have been linked to causal genes[1]. The functional impact of variants that have been linked to causal genes is incompletely understood but is key for the successful integration of genetically-informed patient stratification and precision medicine approaches. Furthermore, GWAS integration has largely focused on associations at the canonical gene level and ignored the complexity of regulation at the transcript level. Importantly, 204 out of 279 confident inflammatory bowel disease (IBD) GWAS genes[2] are regulated by tissue expression quantitative trait loci (eQTL) and tissue splicing quantitative trait loci (sQTL) (Supplementary Fig. 1a). However, the functional impact of isoform-level regulation in IBD is unclear.

The genetic locus that covers the Strawberry notch homologue 2 (*SBNO2*) gene, represents a high confidence association for IBD, specifically Crohn's disease (CD)[3–5] but not ulcerative colitis (UC).

[1]Translational Gastroenterology Unit, University of Oxford, Oxford, UK. [2]Oxford-GSK Institute of Molecular and Computational Medicine (IMCM), Centre for Human Genetics, Nuffield Department of Medicine, University of Oxford, Oxford, UK. [3]Centre for Human Genetics, Nuffield Department of Medicine, University of Oxford, Oxford, UK. [4]Department of Psychiatry, University of Oxford, Oxford, UK. [5]Emory University, Atlanta, USA. [6]Translational Bioinformatics, UCB Pharma, Slough, UK. [7]US Research, UCB Pharma, Durham, NC, USA. [8]Translational Medicine, UCB Pharma, Braine-l'Alleud, Belgium. [9]MRC–Weatherall Institute of Molecular Medicine, University of Oxford, Oxford, UK. [10]Department of Oncology, University of Oxford & Oxford Cancer Centre, Churchill Hospital, Oxford University Hospitals NHS Foundation Trust, Oxford, UK. [11]NIHR Oxford Biomedical Research Centre, Oxford University Hospitals NHS Foundation Trust, Oxford, UK. [12]Department of Paediatrics, University of Oxford, Oxford, UK. [13]Present address: Immunology Disease Area, Novartis Biomedical Research, Basel, CH, Switzerland. [14]Present address: GSK Immunology Network, GSK Medicines Research Center, Stevenage, UK. [15]These authors contributed equally: Isar Nassiri, Suresh Venkateswaran, Subra Kugathasan, Benjamin Fairfax. ✉e-mail: dominik.aschenbrenner@novartis.com; holm.uhlig@ndm.ox.ac.uk

Epigenetic studies demonstrated differential methylation at the *SBNO2* locus in patients with IBD, CD, and UC[6–8]. Besides CD, genotype-phenotype association studies have indicated additional trait associations with genetic variation in *SBNO2*, including heel bone mineral density (HBMD)[9]. The functional link between GWAS and *SBNO2* is not understood. *SBNO2* is a transcriptional regulator of the large DExD/H helicase family and carries a DExD/H-box group and a helicase C-terminal domain. Early studies in *Drosophila melanogaster* identified variants in *sno* (the human *SBNO2* ortholog) that phenotypically resemble those observed in *Notch* mutants[10,11], and revealed *Sno's* regulatory function on gene expression downstream of Notch signalling[10]. Functionally, DExD/H helicase family proteins possess ATP-dependent RNA helicase activity and regulate transcription, mRNA splicing, RNA stability, and RNA metabolism[12,13]. Therefore, SBNO2 may act as a transcriptional regulator through these processes or through direct interaction with transcriptional activators and repressors such as has been described for TBX18[14]. Previous reports highlighted increased *SBNO2* expression downstream of IL-10 signalling[15,16]. In mice, Sbno2 is expressed in various tissues and cell types including astrocytes[17] and immune cells[15]. Deletion of *Sbno2* leads to defective osteoclast differentiation and increased bone mass but no spontaneous inflammatory phenotype[16], despite IL-10's critical role in maintaining immune homoeostasis in mice and humans[18–21].

Here we describe the isoform specific regulation and function of *SBNO2* in human myeloid cells. We demonstrate a role for *SBNO2* as a modulator of myeloid antimicrobial activity and osteocyte transcriptional programmes. We highlight the dependence of *SBNO2*-mediated antimicrobial activity on isoform usage, stimulation context, and host genetics. This provides a functional basis for the observed association of *SBNO2* variants and methylation signals with inflammatory bowel disease, specifically Crohn's disease, and bone development.

## Results

To investigate SBNO2 genotype-phenotype associations[22,23] we plotted GWAS-identified variants[22–25] along genomic coordinates (Fig. 1a). Interestingly, risk variants associated with IBD (and CD in particular) were differentially organised compared to those risk variants that have been linked to HBMD (Fig. 1a). CD specific variants clustered in the proximity of the transcriptional start site (TSS) of the shorter isoform 2 (ENST00000438103, ISO2) of *SBNO2* while those variants associated with CD and HBMD clustered in proximity to the TSS of the longer isoform 1 (ENST00000361757, ISO1) of *SBNO2*. CD only associated variants and those variants associated with CD and HBMD formed two clusters of pairwise linkage disequilibrium (Supplementary Fig. 2) indicating differential inheritance and selection. Structural predictions of human SBNO2 ISO1 suggest an evolutionary conserved helicase-C and an ATPase domain (Supplementary Fig. 3a-c) that are present in both ISO1 and ISO2. The human SBNO2 ISO2 lacks exons at its 5' end, generating a unique 5'UTR and translation initiation, resulting in a 57 amino acid shorter C-terminal end that additionally differs by 36 amino acid residues (Supplementary Fig. 3d). Interestingly, orthologs of human *SBNO2* ISO1 and human *SBNO2* ISO2 can be found in mice, rats, zebrafish, and drosophila indicative of evolutionary conservation (Supplementary Fig. 3e and f). Consistent with SBNO2's role as a transcriptional regulator both SBNO2 ISO1 and ISO2 show intranuclear localization upon ectopic expression in HEK293 cells (Supplementary Fig. 4a–c). Comparing transcriptomic data from terminal ileum biopsy tissues from healthy individuals and patients with CD (Supplementary Table 1) reveals increased *SBNO2* gene expression in inflammation (Supplementary Fig. 5a), while *SBNO2* ISO1 and *SBNO2* ISO2 are the major isoforms expressed (Supplementary Fig. 5b).

To understand the role of SBNO2 in intestinal inflammation and to establish a cellular model for studying SBNO2 function we first identified those human immune cells that express SBNO2 at the protein level. Analysis of mass spectrometry data[26] showed that SBNO2 protein

expression was highest in monocyte-derived dendritic cells (DC) following in vitro activation with TLR-ligands (Supplementary Fig. 6). Furthermore, ENCODE human CD14+ monocytes H3K27Ac, H3K4me3, H3K27me3 and H3k9me3 chromatin immunoprecipitation and sequencing (ChIP-seq) data[27] highlighted the open epigenetic accessibility status of the *SBNO2* locus respective to ISO1 and ISO2 TSS in primary human myeloid cells (Fig. 1a).

To evaluate the activation-dependent regulation of *SBNO2* transcript expression in human myeloid cells, we analysed CD14+ monocytes and monocyte-derived macrophages (MDM) following stimulation with combinations of IL-10, LPS, and IL-10 receptor blocking antibodies (aIL-10R) by RNA-sequencing (RNA-seq) and real time quantitative PCR (RT-qPCR). IL-10 stimulation rapidly induced SBNO2 ISO2 expression but not ISO1 expression, while LPS stimulation induced ISO1 and ISO2 expression (Fig. 1b). Interestingly, transcript-level RNA-seq analysis revealed that *SBNO2* represents the one confident IBD GWAS gene expressed in MDM (238 out of 280 (including SBNO2)) that shows IL-10-specific switching of isoform expression, while several other genes relevant to IBD show changes in transcript expression or usage upon LPS or LPS + aIL-10R stimulation of MDM (Fig. 1c). These include *CD40*, which upon LPS stimulation switches isoform usage from the expression of a stimulatory to an inhibitory receptor in myeloid cells[28]. Furthermore, IL-10 induced greater changes in *SBNO2* isoform expression than IL-6 (Fig. 1d). Blocking IL-10 signalling in the context of LPS stimulation reduced *SBNO2* ISO2 but not *SBNO2* ISO1 expression (Fig. 1d, e) confirming the selective induction of *SBNO2* ISO2 expression downstream of IL-10 signalling (Fig. 1e, f).

Genetic variants determine the quantitative level of gene and isoform expression in a steady state and following stimulation[29,30]. To understand the impact of genetic variation on *SBNO2* gene and isoform expression in myeloid cells, we mapped eQTLs, transcript quantitative trait loci (tQTLs) and isoform quantitative trait loci (isoQTLs) in LPS-stimulated monocytes from healthy Europeans (*n* = 176, Supplementary Table 2), and predicted mechanisms that mediate changes in expression including alteration of the promoter, enhancer, and transcription factor binding site sequence, or 3D genomic interaction (Supplementary Fig. 7a and b, Fig. 2a–c). Strikingly, several CD-associated variants in proximity to the TSS of *SBNO2* ISO2 showed a negative effect on *SBNO2* gene expression (eQTL: rs8178977, rs740495), *SBNO2* ISO2 expression (tQTL: rs4807542, rs2074915, rs2074916) and *SBNO2* ISO2 transcript usage (isoQTL: rs4807542, rs8178977, rs4807569, rs2024092, rs740495, rs2074915, rs2074916; Supplementary Fig. 1), but none affected *SBNO2* ISO1 expression (Fig. 2a–c). rs8178977 represents one example of a CD-associated variant with an *SBNO2* eQTL and an ISO2-specific tQTL, likely mediated by disrupting transcription factor binding at an enhancer region (Fig. 2b, c). While we identified ISO1-specific tQTLs, none were significantly associated with inflammatory disease (Supplementary Fig. 7b). Together these results suggest that CD associated common genetic variation in *SBNO2* is linked to decreased *SBNO2* ISO2 expression and reduced *SBNO2* ISO2 transcript usage in a stimulation-specific context.

To identify clusters of highly correlated genes, and to relate these gene modules to *SBNO2*, ISO1 and ISO2 expression we applied weighted gene correlation network analysis (WGCNA)[31] to RNA-seq data from unstimulated and stimulated MDM (Fig. 3a, Supplementary Fig. 8a–c). These analyses highlighted 17 modules of highly correlated genes across conditions. We identified those modules that showed significant (*p*-value < 0.05) correlation with diverse stimulation conditions, *SBNO2* gene expression (ME6, ME14), *SBNO2* ISO1 transcript expression (ME9, ME15) or *SBNO2* ISO2 (ME12, ME13) transcript expression (Fig. 3a, Supplementary Data 1). MAGMA-based[32] gene-set heritability analysis revealed significant associations of genes in modules ME1, ME5, ME9, and ME15 with IBD and UC, while ME1, ME12, ME13, ME15, and ME17 showed significant associations with CD suggesting

 

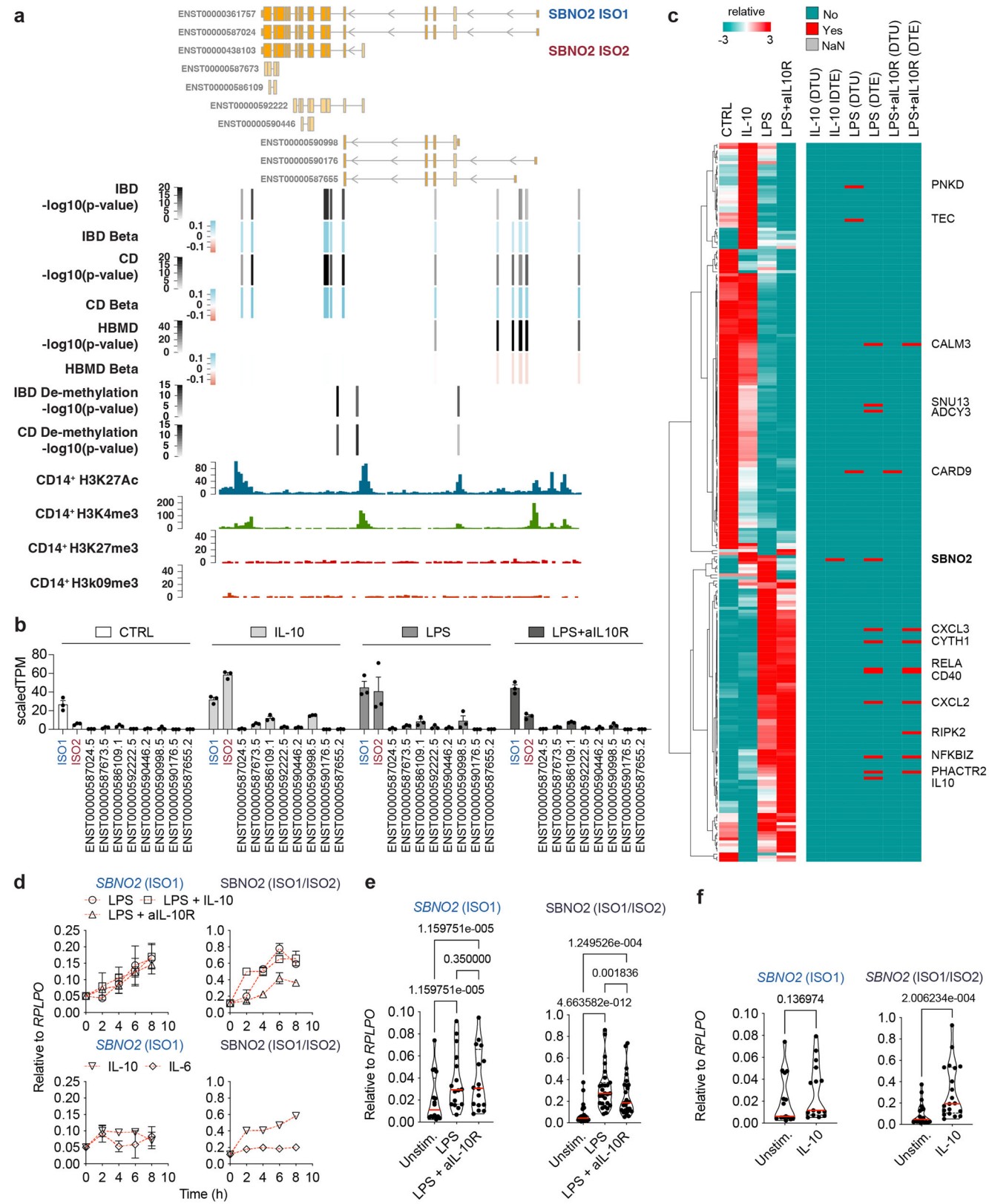

that these modules are relevant for disease development (Fig. 3b, Supplementary Table 3). Interestingly, WGCNA modules ME12 and ME13 that were correlated with IL-10 stimulation in MDM, and that were associated with *SBNO2* ISO2 expression but not *SBNO2* gene or *SBNO2* ISO1 expression, were significantly associated with CD but not IBD or UC. (Fig. 3a). In agreement with the CD-specific association of SBNO2 genetic risk these results show that IL-10 driven *SBNO2* ISO2

expression in MDM is linked to gene expression programmes that specifically associate with CD development.

Next, we assigned biologic functions to those modules that were correlated with *SBNO2* ISO1 expression only (ME9, ME15), *SBNO2* ISO2 expression only (ME12, ME13), and those modules that were correlated with *SBNO2* gene expression (ME6, ME14) (Fig. 3c, Supplementary Fig. 9). Consistent with the identity of WGCNA modules ME9 and ME15

**Fig. 1 | Location, phenotype-genotype association, directionality of genetic variation in *SBNO2*, and differential regulation of *SBNO2* isoform expression. a** Schematic showing the *SBNO2* gene locus, GWAS-based phenotype-genotype association, genomic location, *p*-value (bars and grey colouring), and beta (bars and blue-red colouring) based on the OpenTargets database for variants with significant associations to the *SBNO2* gene locus and with significant associations to IBD and/ or CD and changes in heel bone mineral density (HBMD). Genomic location of IBD-associated and CD-associated demethylation *p*-values at the *SBNO2* locus are indicated[8]. ENCODE CD14+ monocyte histone modification ChIP-seq tracks are aligned to illustrate accessibility and promoter locations in the *SBNO2* locus. **b** Regulation of *SBNO2* transcript expression in human primary MDM following 8 h stimulation with IL-10, LPS, or LPS+aIL10R analysed by RNA-seq (*n* = 3, error bars: SEM). **c** Identification of stimulation-dependent differential transcript expression and usage of confident IBD risk genes within monocyte-derived macrophages. The heatmap shows the row scaled gene expression of those *n* = 238 confident IBD risk genes (out of *n* = 280, including *SBNO2*) that are expressed in monocyte-derived

macrophages in unstimulated (CTRL), IL-10-, LPS-, and LPS + aIL10R-stimulated conditions as identified by RNA-seq. The right map highlights those genes in red that where identified to show differential transcript usage (DTU) or differential transcript expression (DTE) (*p*_adj < 0.05, FDR) in the respective stimulation condition based on staged DRIMseq, DEXseq, and stageR analysis[51]. **d** Kinetics of SBNO2 ISO1 (left) and ISO1/ISO2 (right) expression in CD14+ monocytes following stimulation with IL-6, IL-10, LPS, or LPS+aIL-10R measured by RT-qPCR (*n* = 2, error bars: SEM, 2 independent experiments). **e** Regulation of SBNO2 ISO1 (left) and ISO1/ISO2 (right) expression in MDM following stimulation with LPS, or LPS + aIL-10R analysed by RT-qPCR (ISO1: *n* = 16, 5 independent experiments, ISO1/ISO2: *n* = 27, 10 independent experiments, error bars: SEM, non-parametric, two-sided, Friedman test). **f** Regulation of SBNO2 ISO1 (left) and ISO1/ISO2 (right) expression in MDM following stimulation with IL-10 analysed by RT-qPCR (ISO1: *n* = 14, 5 independent experiments, ISO1/ISO2: *n* = 20, 7 independent experiments, non-parametric, two-sided, Mann-Whitney test) shown as relative expression. Source data are provided as a Source Data file for Fig. 1b–f.

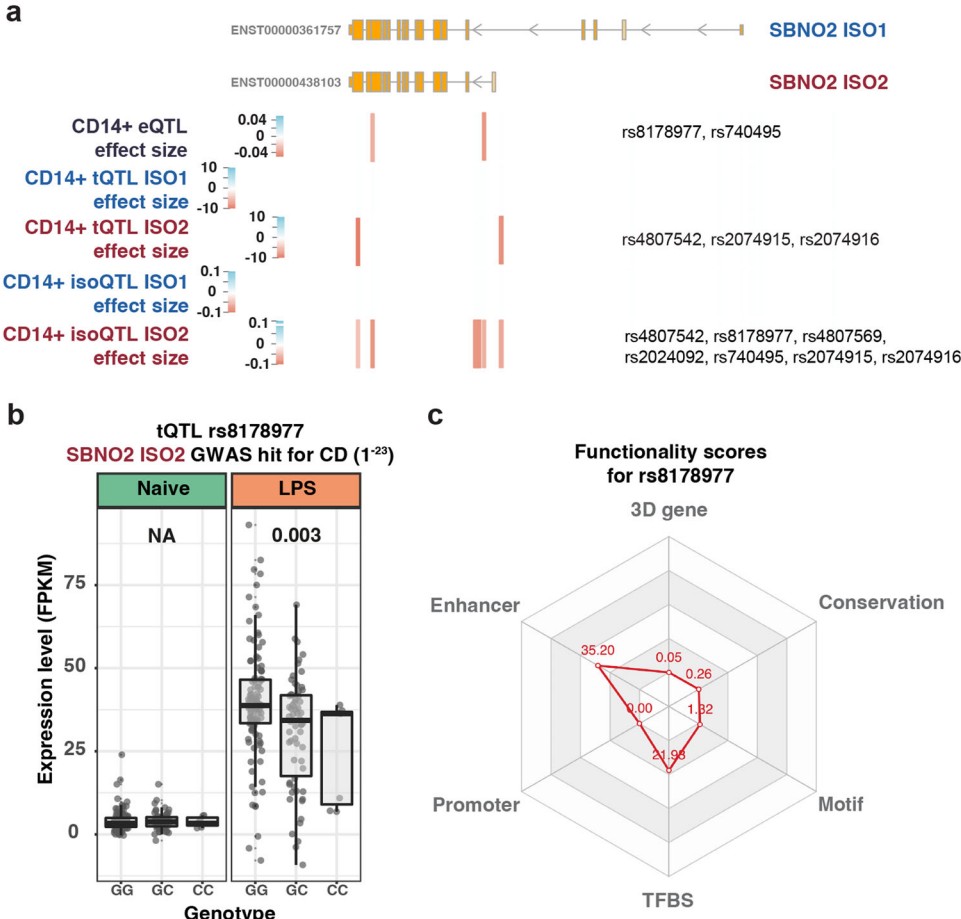

**Fig. 2 | *SBNO2* gene expression and transcript usage is regulated by genetic variation. a** Schematic showing the *SBNO2* gene locus, location of variants with GWAS-based association with IBD/CD in *SBNO2*, and respective location and effect size of *SBNO2* ISO1 and ISO2 tQTLs, and ISO1 and ISO2 isoQTLs in LPS-stimulated CD14+ monocytes. **b** Example of an IBD-associated variant (rs8178977) with a tQTL specifically affecting ISO2 expression in LPS-stimulated CD14+ monocytes. Box plots depict the interquartile range as the lower and upper bounds, respectively. The whiskers represent minimum and maximum, and the centre depicts the median. eQTL analysis was performed with FastQTL and QTLtools using a linear

regression. To allow comparison with output of the regression model the optimal number of PC was used to regress out expression changes attributable to the effect of the non-genetic covariates in local association plots. **c** 3D SNP score was applied to evaluate the functional significance of rs8178977 in 6 categories including 3D interacting genes, enhancer state, promoter state, transcription factor binding sites, sequence motifs altered, and conservation categories. The score for a SNP is calculated using the number of hits in each functional category in human monocytes from the GTEx project and a Poisson distribution model. Source data are provided as a Source Data file for Fig. 2b.

highly correlated (module correlation > 0.5) member genes (e.g. *HLA, IFITM2, IFIT1, IFIT3, IFNAR2, IFNGR2, DAP, STAT1, STAT2, STAT5A, CCL3, CCL5, CCL18,NFKB1, NFKB2, MYD88, IRF1, IRF2, IRF7, IRF9, OAS1, OAS2, OAS3, IFIH1, IFI35, RELB*; Supplementary Data 1), *SBNO2* ISO1-expression-correlated modules were linked to pathways of

inflammatory responses (e.g. cytokine and interferon signalling). *SBNO2* ISO2 expression, that was correlated with ME12 and ME13 member genes (e.g. *BCL2L1, CTSL, PIK3CD, PTEN, BNIP3L, CAPN1, CAPNS1, GAPDH, GNAI3, HSPB1, SREBF1, GPR137B, ATP13A2, RAB3GAP2, SNX6, LGALS8, RAB7A, AP3D1, CSNK1D*) associated with pathways

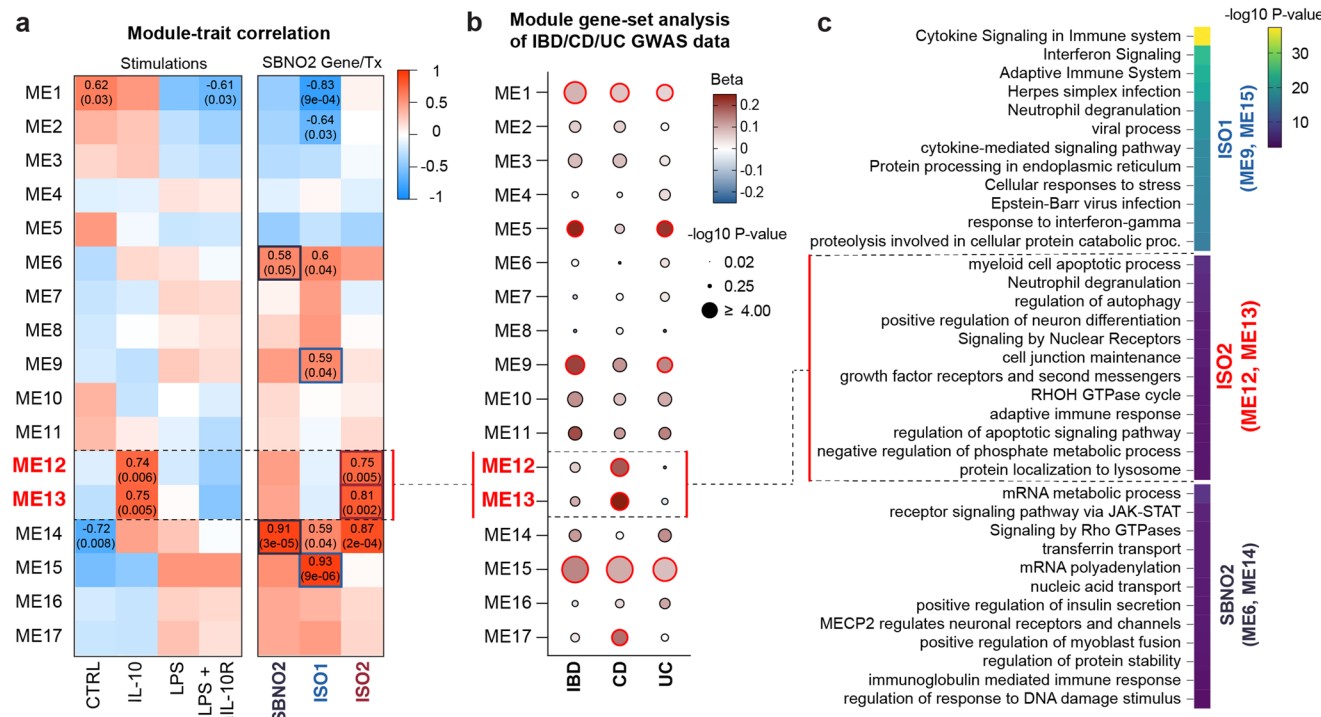

**Fig. 3 | Definition of functional gene modules associated with *SBNO2* gene and isoform expression in human primary MDM. a** Correlation of the 17 identified WGCNA modules of genes (*y*-axis) with stimulations, genes, and transcripts of interest (x-axis). Correlation coefficients and *p*-values are indicated for modules with *p*-values ≤ 0.05 (Pearson correlation, two-sided test, uncorrected *p*-values). **b** MAGMA-based[32] gene-set heritability analysis on WGCNA modules for IBD, CD, and UC[65]. The dot blot illustrates the directionality of association as colour code and the enrichment significance (-log10 *p*-value) as dot size. Red circular borders around individual dots indicate *p*-values < 0.05 (one-sided test, uncorrected *p*-values, as implemented in MAGMA[32]). **c** Enrichment of GO pathway terms based on those gene modules that were found correlated with ISO1 (ME9 and ME15), ISO2 (ME12 and ME13), and ISO1 and ISO2 (ME6 and ME14) expression (hypergeometric test, Bonferroni correction, $p_{adj}$ < 0.05). Source data are provided as a Source Data file for (**b**, **c**).

linked to antimicrobial activity (e.g. autophagy, lysosome) (Fig. 3c and Supplementary Data 1). *SBNO2* gene expression was correlated with mRNA metabolic processes, JAK-STAT signalling, and GTPase signalling (Fig. 3c).

To understand SBNO2's impact on MDM function in more detail we performed siRNA-mediated knockdown and RNA-seq (Fig. 4a, b, Supplementary Fig. 10a-d). Interestingly, we found known IL-10-signalling suppressed genes downregulated upon knockdown of *SBNO2* (e.g. *IL23A*, *IL20*, and *IL24*[33], Fig. 4a, b). Furthermore, the expression of chemokine receptors *CXCR2* and *CXCR4*, that influence myeloid cells tissue homing and function[34,35] were negatively regulated upon *SBNO2* knockdown. Several genes involved in the regulation of osteoblast and osteoclast function, including *KREMEN1*, a regulator of bone formation[36,37], that was found upregulated upon *SBNO2* knockdown (Fig. 4b), were found deregulated upon *SBNO2* knockdown (Upregulated: *CSF1R, CYP27B1, FAM20C, GREM1, LMNA, LRP5, SORT1, SPP1, TNEN119,* and *TNS3*; Downregulated: *THBS1, ACP5, BMP6, CA2, DUSP4, INHBA, NFIL3,* and *VDR;* Supplementary Data 2). These results indicated that *SBNO2* may not mediate the conventional anti-inflammatory effects of IL-10.

To identify perturbed functional pathways, we performed an enrichment analysis across stimulation conditions and those genes that were found differentially regulated by *SBNO2* knockdown (Fig. 4c–g, Supplementary Data 3-6). We identified condition-specific and shared pathways upon LPS stimulation in the presence or absence of *SBNO2* ISO2-inducing IL-10 signalling (Fig. 4c). In LPS- and LPS + aIL-10R-stimulated MDM *SBNO2* knockdown-downregulated genes (SBNO2-induced genes) contained *THBS1* (Thrombospondin 1, TSP1) as top regulated gene (Fig. 4d, g). Functionally, TSP1 has been linked to bone development, normal lung homoeostasis[38], and synaptogenesis in

astrocytes[39]. At the pathway-level *SBNO2* knockdown-downregulated genes in LPS- and LPS+aIL-10R-stimulated MDM associated with the term "p53 signalling pathway". Strikingly, in the context of LPS but not LPS + aIL-10R stimulation SBNO2-induced genes (e.g. *ATP6V1A, ATP6V1B2, ATP6V1D, ATP6V1F, ATP6V1G1, ATP6V1H, ATP6V1E1* and *2, ATP6V0A1* and *2, ATP6V0B, ATP6V0C, ATP6V0D1* and *2, ATP6V0E1* and *2, ATP6AP1, TCIRG1, MCOLN1, LMTK2, ARHGAP1, RAB11B, TFRC, DNM2, STEAP3, ATP6V1C1, TF, HFE, CLTC*) associated with the term "Transferrin transport" as the top regulated pathway (Fig. 4d, g), suggesting that IL-10-induced *SBNO2* expression promoted ATPase function, that is a critical mechanisms in endocytosis, lysosomal function, phagosome acidification, and bone resorption[40]. Interestingly, SBNO2 has previously been identified as a regulator of autophagy-dependent intra-cellular pathogen defence in a GWAS-based IBD genes siRNA screen[41]. In contrast, SBNO2-suppressed genes and pathways largely overlapped between LPS and LPS+aIL-10R conditions (Fig. 4e) and included the regulation of pattern recognition receptor (PRR) signalling (Fig. 4e) by direct negative regulation of PRRs *TLR1, TLR2, TLR4, TLR6* and *NOD2* expression (Fig. 4g). LPS+aIL-10R stimulation-regulated genes pathway term enrichment that was not shared with LPS stimulation included terms linked to complement activation and included genes such as *C2, C3, C4A, CFB* and *CFP* (Fig. 4e, g). In addition, *SBNO2* knockdown deregulated LPS-specific, LPS+aIL-10R-specific, and shared genes associated with functional pathways of the acute phase response (SBNO2-induced: *STAT3, HAMP, SAA1, SAA2, LBP*; SBNO2-suppressed: *IL1A, FN1, IL1B, ITIH4, IL6*), signal transducer and activator of transcription (STAT)-dependent cytokine signalling (SBNO2-induced: *PECAM1, LYN, IL15, JAK2, CSFR1*; SBNO2-suppressed: *IL20, CSF2, IL23A, IL24, LIF*), and che-mokine secretion and sensing respectively (SBNO2-induced: *CXCL9, CXCL13, CCL1*; SBNO2-suppressed: *CXCR2, CXCR4, CCL22*) (Fig. 4f, g).

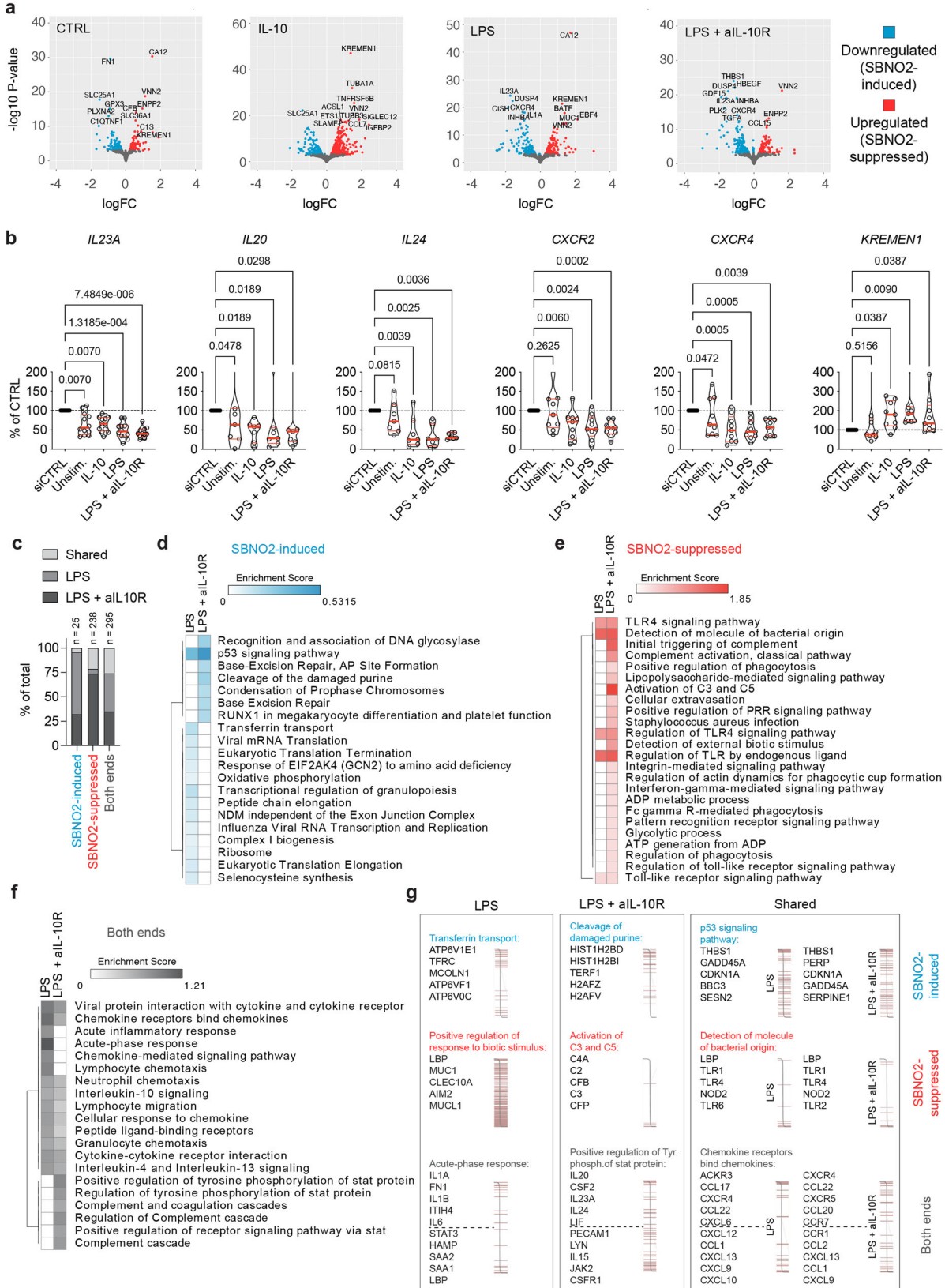

Together these analyses suggest a function for IL-10-induced *SBNO2* ISO2 in the regulation of myeloid antimicrobial activity by influencing ATPase function and lysosome maturation, while both SBNO2 isoforms regulate gene programmes related to microbial recognition, stimulation-induced inflammatory responses, and bone homoeostasis.

These results prompted us to explore antimicrobial cellular functions in myeloid cells, a critical mononuclear phagocyte function that can lead to intestinal inflammation in humans when impaired[21]. To test this, we performed siRNA-mediated knockdown of *SBNO2* in primary human MDM and analysed antimicrobial activity by gentamycin protection assay after *Salmonella Typhimurium* infection (Fig. 5a, b).

**Fig. 4 | siRNA-mediated knockdown of *SBNO2* in primary human MDM.**
**a** Volcano plots illustrate differential gene expression following siRNA-mediated *SBNO2* knockdown in unstimulated MDM and MDM following 8 h stimulation with IL-10, LPS, or LPS + aIL-10R analysed by RNA-seq and DESeq2 ($n = 3$, log2 fold change (log$_2$fc) > 0.5, $p_{adj}$ < 0.05, FDR). **b** RT-qPCR validation of target genes of interest found differentially regulated by *SBNO2* knockdown in RNA-seq experiments (Independent experiments/donors: *IL23A*: $n = 6/12$; *IL20, IL24*: $n = 3/7$; *CXCR2, CXCR4, KREMEN1*: $n = 4/9$; non-parametric, two-sided, Friedman test). **c** The bar graph illustrates the overlap of functional pathways (GO Biologic Processes, KEGG pathways, and Reactome Pathways) based on STRING database functional

enrichment analysis of SBNO2 knockdown differentially regulated genes in LPS- and LPS + aIL-10R-stimulated MDM. The total number of enriched pathways is indicated. **d–f** The heatmap shows the enrichment score of STRING database functional enrichment analysis showing the (**e**) top 10 downregulated/SBNO2-induced pathways, (**f**) top 10 upregulated/SBNO2-suppressed pathways, and top 10 enriched pathways without polarity in the dataset based on DESeq2 log$_2$ fold change differential expression ranking. **g** Examples of enriched functional pathway terms and list of respective top 5 SBNO2-regulated pathway genes (enrichment score). The distribution according to DESeq2 log$_2$ fold change expression across the data are visualized. Source data are provided as a Source Data file for Fig. 4b.

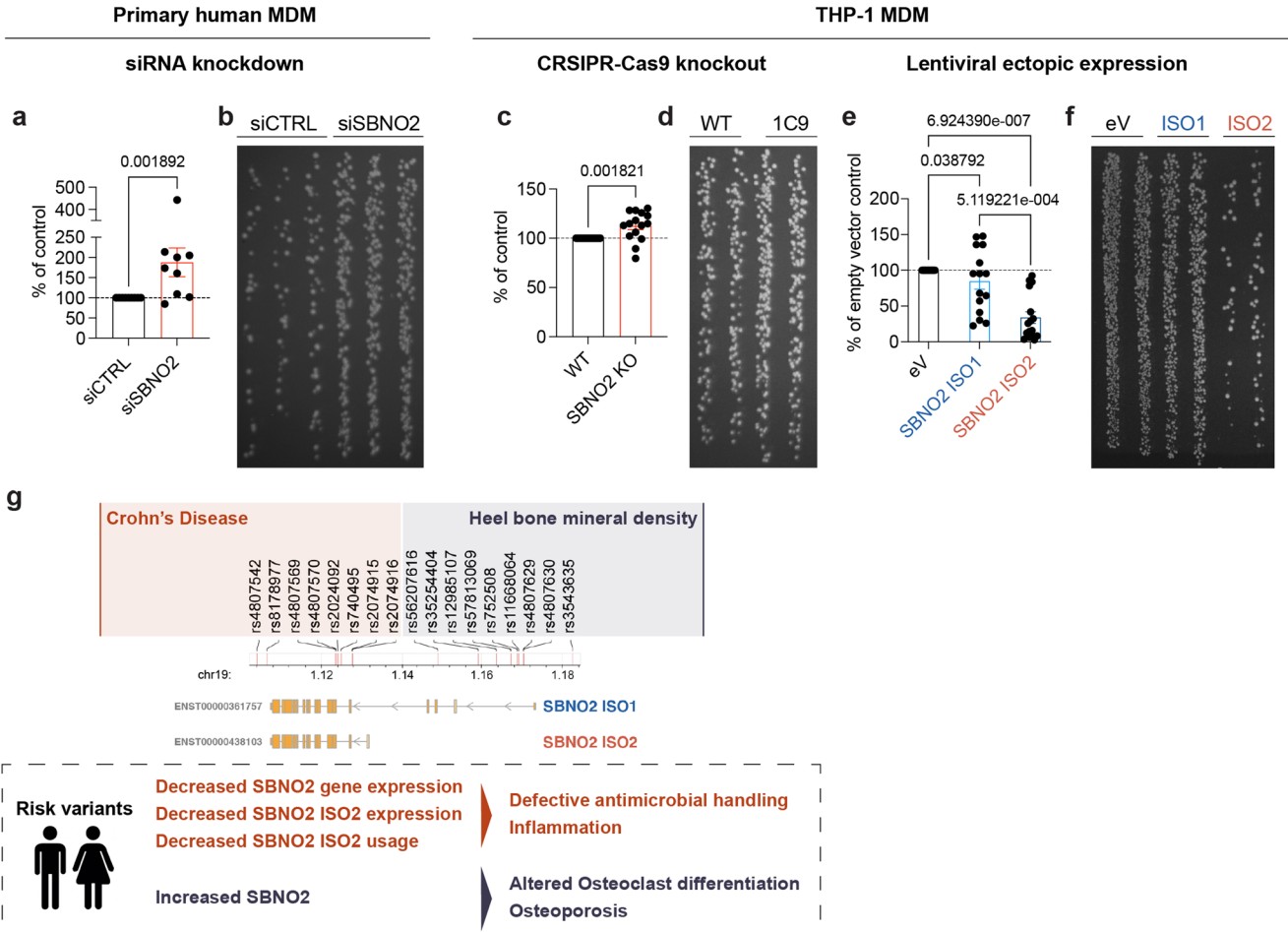

**Fig. 5 | Modulation of *SBNO2* expression regulates intracellular bacterial killing in primary human MDM and THP-1 MDM. a** Gentamycin protection assay (GPA) in primary MCSF-differentiated human MDM upon siRNA-mediated SBNO2 knockdown using *Salmonella typhimurium* infection. The bar graph shows *S. typhimurium* colonies from lysed siSBNO2-treated MDM following overnight culture as % of colonies from lysed control siRNA-treated MDM ($n = 9$, 3 independent experiments, error bars: Mean with SEM, non-parametric, two-sided, Mann-Whitney test).
**b** Representative LB-agar image showing *S. typhimurium* colonies from lysed primary MDM following overnight culture (dilution 1:10) according to (**a**). **c** GPA in PMA-differentiated WT THP-1 and CRSPR-Cas9 knockout THP-1 (single cell clone 1C9) PMA-differentiated MDM ($n = 15$, 3 independent experiments, error bars: Mean with SD, non-parametric, two-sided, Mann-Whitney test). **d** Representative LB-agar

image showing *S. typhimurium* colonies from lysed THP-1 MDM following overnight culture (dilution 1:100) according to (**c**). **e** GPA in PMA-differentiated THP-1 MDM upon ectopic expression of SBNO2 isoforms ISO1 and ISO2. Results are shown relative to the number of *S. typhimurium* colonies obtained from lysed empty vector (eV) expressing THP-1 MDM ($n = 15$, 6 independent experiments, error bars: Mean with SD, non-parametric, two-sided, Kruskal-Wallis test). **f** Representative LB-agar image showing *S. typhimurium* colonies from lysed THP-1 MDM following overnight culture (dilution 1:10) according to (**e**). **g** Graphical summary of the model for SBNO2 isoform-specific disease risk and anti-microbial activity in Crohn's disease, and altered osteoclast differentiation in osteoporosis. Source data are provided as a Source Data file for Fig. 5a–f.

Indeed, knockdown of SBNO2 caused reduced intracellular killing of *S. typhimurium*. In agreement, CRISPR-Cas9 mediated knockout of SBNO2 in THP-1 derived macrophages impaired antibacterial activity (Fig. 5c, d). Furthermore, ectopic expression of SBNO2 ISO1 and in particular SBNO2 ISO2 increased killing of intracellular *S. Typhimurium* (Fig. 5e, f, Supplementary Fig. 11a–c) Corresponding with observations

in SBNO2 knockout mice[16], ectopic expression of both SBNO2 ISO1 and SBNO2 ISO2 regulated genes related to osteoblast and osteoclast differentiation, including *OCSTAMP* and *DCSTAMP* (Supplementary Fig. 11d, e). In conclusion, we demonstrate that differential *SBNO2* isoform expression is dependent on stimulation context and two clusters of genetic variants mediating distinct functions and risk associated with

human disease. In alignment with this hypothesis, biallelic loss of function variants that affect the catalytic core and both isoforms of SBNO2 cause immunodeficiency and bone malformations[42].

## Discussion

To unravel the mechanistic complexity of multigenic disorders such as IBD[43] represents a prerequisite for personalized medicine where those patients that are most likely to respond receive disease- and context-specific treatment. Patient care in rare monogenic diseases aims for precision medicine and studies focusing on the elucidation of the pathogenesis and mechanistic basis continuously inform key checkpoints of the immune system and enhance our understanding of health and disease, also in IBD[21]. Additionally, genetically informed models provide potential for accurate definition of novel drug targets and identification of key pathways for successful therapeutic interventions. Still, the mechanistic basis for genetic associations in immune mediated diseases, including classical IBD, remains to be fully elucidated. Furthermore, previous studies have preferentially focused on understanding genetic disease risk at the canonical gene level and demonstrated the link between GWAS signals and eQTLs, and the importance of cell-type or cell-state specific regulation, but largely omitted the complexity of transcript level regulation. The distinct role of sQTLs in the genetic regulation of transcription and complex trait variation has only been appreciated more recently[44–48].

In this study we investigated the genetic and transcriptional regulation of the GWAS-identified IBD risk gene SBNO2 and the related gene network in human monocytes and macrophages. Our analysis revealed differential expression of SBNO2 isoforms in dependence of exogenous IL-10 stimulation and also LPS-induced autocrine IL-10 signalling that specifically increased SBNO2 ISO2 expression. We showed that 15 high confidence IBD risk genes changed transcript usage or expression in LPS or LPS + aIL-10R stimulated MDM, while SBNO2 represented the one high confidence IBD risk gene expressed in MDM that changed transcript usage and transcript expression upon IL-10 stimulation. In addition, we demonstrated the induction of SBNO2 ISO2 expression downstream of IL-10 but not IL-6 signalling, suggestive of the presence of a yet unidentified mechanism of selectivity that discriminates anti-inflammatory (IL-10-dependent) from pro-inflammatory (IL-6-dependent) STAT3 signalling at the target gene locus, adding an additional layer of control to the regulation of SBNO2 gene expression.

Based on the analysis of genetic regulatory effects in 176 healthy donors steady state and LPS-stimulated monocytes we showed reduced SBNO2 expression, SBNO2 ISO2 expression, and SBNO2 ISO2 transcript usage upon LPS stimulation, providing a mechanistic basis for the association of genetic variation at the SBNO2 locus with Crohn's disease. Furthermore, combination of WGCNA[31] and MAGAMA[32] analyses highlighted the specific association of an IL-10-driven SBNO2 ISO2-specific gene network with CD heritability that was functionally associated with pathways relevant for bacterial handling. Although we initially hypothesized that LOF in SBNO2 would lead to de-regulated inflammatory responses by disrupting IL-10-dependent suppressive mechanisms we could not confirm this assumption. Instead functional characterization of MDM bacterial handling capacity in gentamycin protection assays using ectopic expression and CRISPR-Cas9 knockout in THP-1 MDM point towards a causal role of immunodeficiency due to decreased anti-microbial activity in the development of intestinal inflammation in patients with GWAS-associated genetic variants that are linked to decreased expression or usage of SBNO2 ISO2. On this basis it is tempting to speculate that epigenetic regulation and disease-associated genetic variation at the SBNO2 locus may impact on the prevalence of SBNO2 isoform transcription, and ultimately impact on cellular function. Furthermore, these results may also explain the immunodeficiency and infection susceptibility observed in patients with biallelic LOF in SBNO2[42].

In contrast to SBNO2 ISO2, SBNO2 ISO1 expression did not depend on IL-10 signalling but on LPS stimulation. Variants located in proximity to the SBNO2 ISO1 TSS associated with changes in HBMD, which may relate to SBNO2's role in regulating osteoblast and osteoclast function. Indeed, increased expression of SBNO2 has been found in male ankylosing spondylitis associated osteoporosis[49]. Previous studies performed in Sbno2 knockout mice have revealed a regulatory function in osteoclast differentiation[16] and knockdown and ectopic expression experiments presented in this study support a role for SBNO2 in the regulation of human osteoclast multinucleation and differentiation and support evolutionary conserved functions of SBNO2 in normal bone development. One of the top SBNO2 positively regulated genes across stimulation conditions was OCSTAMP, a gene coding for a surface receptor involved in osteoclast differentiation with biologic functions similar to DC-STAMP[50,51], that was found regulated by SBNO2 in the murine gene knockout model[16]. The involvement of SBNO2 in the positive and negative regulation of osteoclast and osteoblast function respectively suggests a role in maintaining the optimal balance of osteoclast versus osteoblast differentiation transcriptional programmes that orchestrate bone reduction versus formation and may explain the presence of retained teeth in patients with rare disease caused by LOF in SBNO2[42].

While we have focused our analysis on monocytes and macrophages, this study also highlights potential roles of SBNO2 in the differentiation and function of other cell types, in particular neutrophils. Indeed, the impact of SBNO2 on the expression of CXCR2 and CXCR4, and genes involved in the oxidative stress response (e.g. SOD2, GPX3) suggests an additional role in the regulation of biologic processes that are key to normal neutrophil function. Normal CXCR2 and CXCR4 expression in neutrophils is required for the optimal maturation and release of neutrophils from bone marrow into circulation[52]. An impairment of CXCR2 signalling in neutrophils would likely cause the premature release of dysfunctional neutrophils into circulation[52], a testable hypothesis that can be adressed in the context of rare human disease. In addition, impaired oxidative stress response in neutrophils would lead to reduced elimination of radical oxygen species thereby allowing increased damage to host tissue during an immune response. Furthermore, the decreased production of hypochlorous acid from superoxide would additionally reduce the antibacterial activity of neutrophils[53].

In conclusion, we demonstrate that SBNO2 regulates key cellular properties of human myeloid cells in inflammation, anti-microbial response, and osteoclast differentiation and provide a mechanistic basis for the identified association of human genetic variation at the SBNO2 locus with increased risk for the development of chronic intestinal inflammation and reduced bone density.

## Methods

### Ethics Approval
Experiments were carried out with Research Ethics Board (REB) approval from the Oxford IBD cohort study (Rare disease subproject). Informed written consent to participate in research was obtained from patients/families and controls.

### Cell culture and stimulation
Peripheral blood mononuclear cells (PBMCs) were isolated by gradient Ficoll centrifugation (Sigma-Aldrich). CD14+ monocytes were isolated by positive selection using magnet-assisted cell sorting (MACS) and CD14+ MicroBeads (Miltenyi; Cat.#130-050-201) according to the manufacturer's instructions. CD14+ monocytes were cultured in RPMI-1640 (Sigma) supplemented with 10% foetal calve serum (FCS, Sigma), 100 U/mL penicillin, and 10 μg/mL streptomycin (Sigma), hereafter called R10. Monocyte-derived macrophages (MDM) were generated by culturing $5 \times 10^6$ CD14+ monocytes in 10 cm cell culture dishes (Corning) at a cell density of $0.5 \times 10^6$ cells/mL in R10 supplemented with

100 ng/mL human recombinant M-CSF (Peprotech; Cat.#216-MC) for 5 days. CD14⁺ monocytes or MDM were stimulated for indicated time points individually or with combinations of 200 ng/mL LPS (Enzo Life Sciences; Cat.#ALX-581-008), 100 ng/mL recombinant human IL-10 (Peprotech; Cat.#200-10), recombinant human IL-6 (Peprotech; Cat.#200-06), and 10 µg/mL IL-10R blocking antibodies (Biolegend; Cat.#308818; clone: 3F9) in R10.

### siRNA-mediated knockdown in primary human MDM

Day 5 M-CSF-differentiated MDM (see Cell culture and stimulation) were detached from 10 cm dishes (Corning) using 5 min incubation in Versene Solution (Gibco) and manual scraping. MDM were plated at a cell density of $0.5 \times 10^5$ cells/well in 24-well plates (Corning) and incubated overnight in 1 mL R10 supplemented with 100 ng/mL human recombinant M-CSF (Peprotech; Cat.#216-MC). On the following day the medium was replaced with 125 µL phenol red free RPMI-1640 (Gibco). DharmaFECT transfection reagent (2.5 µL/well; Horizon; Cat.#T-2002-03), serum-free OptiMEM (Gibco), and 100 nM ON-TARGETplus Human SBNO2 siRNA SMARTpool (Horizon; Cat.#L-004743-01-0020) or 100 nM ON-TARGETplus Non-targeting Control (Horizon; Cat.# D-001810-10-20) were prepared according to the manufacturer's instructions and added to the respective cultures. Following 2 hrs incubation at 37 °C the transfection complexes were removed and replaced with 0.5 mL phenol red free RPMI-1640 (Gibco) supplemented with 10% FCS (Sigma) and 100 ng/mL human recombinant M-CSF (Peprotech; Cat.#216-MC). Following 48 hrs incubation cells were used for stimulation experiments.

### Generation of THP-1 MDM

THP-1 MDM were generated by culturing $3 \times 10^6$ THP-1 monocytes in 10 cm cell culture dishes (Corning) at a cell density of $0.5 \times 10^6$ cells/mL in R10 supplemented 100 ng/mL Phorbol 12-myristate 13-acetate (PMA, Sigma-Aldrich; Cat.#P8139) for 12 hrs. After 12 hrs PMA was washed out by carefully adding and aspirating 10 mL of RPMI-1640 (Gibco) to adherent THP-1 cells, two times. Following, 6 mL of R10 were added and incubation continued for 48 h. THP-1 source: ATCC; Cat.# TIB-202.

### Immunofluorescence

HEK293 cells (ATCC) were seeded on tissue culture slides (Sarstedt). 24 hrs later, cells were transfected with combinations of plasmids coding for Myc-tagged SBNO2 ISO1 (Genscript), HA-tagged SBNO2 ISO2 (Genscript), and a GFP-coding plasmid using Lipofectamin2000 (Invitrogen). Cells were fixed for 30 min at 37 °C in 3.7% paraformaldehyde (Sigma), permeabilized using -20 °C MeOH (Sigma) treatment for 30 min on ice. Immunofluorescence staining was for 1 hr at RT using anti-c-Myc antibody (Biolgend, Cat.# 626810, clone 9E10, dilution: 1:100) and anti-HA antibody (Cell Signalling, Cat #55420, clone C29F4, dilution: 1:100). Cells were counterstained with DAPI and sealed with vecta shield (Vector labs). Images were acquired on a Zeiss LSM 880 microscope using a 40X objective lens and 63X/1.4 oil objective lens and the ZEN2011 software. HEK293 source: ATCC; Cat.# CRL-1573.

### Lentiviral production and transduction

The empty vector pCDH-EF1-MCS-T2A-copGFP (CD526A-1) and the vector carrying human SBNO2 transcript variant 1 (NM_014963.3) or SBNO2 transcript variant 2 (NM_001100122.2) for ectopic expression were purchased from SBI Systems Biosciences. Lentiviral particles were produced by transiently transfecting HEK293T (ATCC) cells with the above described vectors together with the ViraPower™ lentiviral packaging mix (Invitrogen) in 150 mm cell culture dishes (Corning). Briefly, HEK293T cells were transfected with a cocktail of transfer vector and packaging mix in Opti-MEM (Gibco), using Lipofectamine® 2000 (Thermo Fisher, Cat.# 11668019) as transfecting agent according to the manufacturer's instructions. Culture supernatants containing

viral particles were harvested at 72 h post-transfection and titre were determined by limiting dilution on HEK293 cells. THP-1 lines were transduced by spin infection (90 min; 800 g; 32 °C) in the presence of 5 µg/mL Polybrene (Sigma, Cat.# TR-1003-G). The medium was then replaced. Following expansion, GFP⁺ cells were FACS-sorted on a BD FACSAria III using the BD FACSDiva v9.0, to establish pure GFP⁺ THP-1 cell lines.

### Gentamycin protections assay

Infection experiments were performed as previously described[54,55]. Primary MDM were infected at 1:10 MOI and THP-1 MDM were infected at 1:100 MOI with GFP-*Salmonella Typhimurium* (NCTC 12023) for 1 h. Cells were then treated with 100 µg/mL gentamicin (Sigma) and subsequently lysed with 1% Triton X-100 (Sigma). Lysates were then plated on LB-agar plates (Sigma) using the track method.

### RNA extraction, cDNA synthesis and real-time RT-qPCR

Prior to total RNA-extraction cells were washed once in PBS (Sigma). Total RNA was extracted using the Omega BIO-TEK E.Z.N.A.® Total RNA Kit I (Omega BIO-TEK; Cat.# R6834-02) according to the manufacturer's instructions. DNA digestion was performed on-column using the Qiagen RNase-Free DNase Set (Qiagen; Cat.# 79256). cDNA was generated from 1 µg RNA (assessed by Nanodrop2000 (Thermo Fisher Scientific) measurement) using the Applied Biosystems High-Capacity cDNA Reverse Transcription Kit (Applied Biosystems; Cat.#4368814). Real-time qPCRs were performed in 96-well plates using the PrecisionPLUS qPCR Mater Mix (Primer Design; Cat.#4368814) and the CFX96 Touch Real-Time PCR Detection System machine (BIO-RAD) and BIO-RAD FX96 Touch Real-Time PCR Detection System - CFX Maestro Software. The expression of transcripts was normalized to expression of Large Ribosomal Protein (RPLP0). Data analysis was performed using the Cycles threshold (ddCt) method and expressed as mRNA relative expression ddCt. The following TaqMan probes (Applied Biosystems) were used for qPCR analysis: RPLP0 (Hs99999902_m1), SBNO2 (ISO1/ISO2) (Hs00922127_m1), SBNO2 (ISO1) (Hs00209130_m1), IL20 (Hs00218888_m1), IL23A (Hs00900828_g1), IL24 (Hs01114274_m1), CXCR2 (Hs00174304_m1), CXCR4 (Hs00976734_m1), KREMEN1 (Hs00230750_m1)

### RNA-sequencing

Total RNA quantity and quality were assessed using an Agilent 4200 TapeStation System and the RNA ScreenTape Analysis kit (Agilent). Only samples with a RIN ≥ 9.4 were processed. Library preparation, clustering and sequencing were performed by Novogene. A total amount of 1 µg RNA per sample was used as input material for the RNA sample preparations. Sequencing libraries were generated using NEBNext® UltraTM RNA Library Prep Kit for Illumina® (New England Biolabs (NEB)) following manufacturer's recommendations and index codes were added to attribute sequences to each sample. Briefly, mRNA was purified from total RNA using poly-T oligo-attached magnetic beads. Fragmentation was carried out using divalent cations under elevated temperature in NEBNext First Strand Synthesis Reaction Buffer (5X). First strand cDNA was synthesized using random hexamer primer and M-MuLV Reverse Transcriptase (RNase H). Second strand cDNA synthesis was subsequently performed using DNA Polymerase I and RNase H. Remaining overhangs were converted into blunt ends via exonuclease/polymerase activities. After adenylation of 3′ ends of DNA fragments, NEBNext Adaptor with hairpin loop structure were ligated to prepare for hybridization. In order to select cDNA fragments of preferentially 150 ~ 200 bp in length, the library fragments were purified with AMPure XP system (Beckman Coulter, Beverly). Then 3 µl USER Enzyme (NEB) was used with size-selected, adaptorligated cDNA at 37 °C for 15 min followed by 5 min at 95 °C before PCR. Then PCR was performed with Phusion High-Fidelity DNA polymerase, Universal PCR primers and Index Primer. At last, PCR

products were purified (AMPure XP system) and library quality was assessed on the Agilent Bioanalyzer 2100 system. The clustering of the index-coded samples was performed on a cBot Cluster Generation System using PE Cluster Kit cBot-HS (Illumina) according to the manufacturer's instructions. After cluster generation, the library preparations were sequenced on an Illumina platform and 150 bp paired-end reads were generated with $30 \times 10^6$ reads/sample on average.

## RNA-sequencing analysis

Pre-processing of RNA-seq data was performed by Novogene. Raw reads were filtered to remove reads with adaptor contamination or reads with low quality. Only clean reads were used in the downstream analyses. Removed reads included reads with adaptor contamination, reads when uncertain nucleotides constitute >10% of either read ($N > 10\%$), and reads with low quality nucleotides (base quality < 20) i.e. when these constituted >50% of the read. Files were aligned to hg38 using hisat2[56]. Transcript quantification was performed using Salmon[57] version 1.2.1. Data were imported into R (version 3.6.3) and R studio (version 1.1.456) as "scaledTPM" with tximport[58]. Transcript to gene mapping was performed using AnnotationDbi (version 1.48). Genes that did not show at least 10 counts in two samples were excluded from analysis. Differentially expressed genes were identified by DESeq2[59] using the apeglm[60] method (log2 fold change ($log_2fc$) > 0.5, $p_{adj} < 0.05$, FDR). Volcano plots were generated with ggplot2 (version 3.3.1)[61] and heatmaps using the pheatmap package (version 1.0.12). Gene ontology analysis was performed using the STRING Functional Enrichment Analysis[62].

## Analysis of differential transcript expression and transcript usage in monocyte-derived macrophages

Detection of differential transcript expression (DTE) and differential transcript usage (DTU) from RNA-seq data was performed according to the previously described bioinformatic workflow[63]. Briefly, transcript quantification was performed using Salmon[57] version 1.2.1. Data were imported into R (version 3.6.3) and R studio (version 1.1.456) as "scaledTPM" with tximport[58]. Transcript to gene mapping and generation of a TxDb object were performed using the GenomicFeatures package (version 1.38.0). Differential transcript usage/expression was analyzed using the DRIMseq package (1.14.0). Only transcripts with a count of at least 10 in at least 50% of samples, a relative abundance proportion of at least 0.1 in at least 50% of samples, and a total count of the corresponding gene of at least 10 in all samples were analyzed. The stageR package (1.8.0) with an alpha = 0.05 was used for two stage testing and to determine DTU/DTE. The DEXseq package (1.32.0) was then used on the DRIMseq-filtered dataset to define DTU/DTE. The stageR package (1.8.0) with an alpha = 0.05 was then used to generate a table with overall false discovery rate (OFDR) control and FDR-adjusted $p$-values.

## WGCNA analysis

Scaled TPM values from RNA-seq analysis were imported into R (version 3.6.3) and R studio (version 1.1.456). Non-expressed transcripts were removed, the remaining data upper-quartile normalised and $log_2(n + 1)$ transformed. Prior to WGCNA analysis the dataset was filtered to retain genes that had a transformed expression value ≥ 0.1 in >5% of samples. The WGCNA R package (version 1.69)[31] was then applied. Data were cleaned using the "goodSamplesGenes" function (with parameters: minFraction = 0.4, minNsamples = 4, minNGenes = 4, minRelativeWeight = 0.1). Adjacencies were calculated with the "adjacency" function (parameters: type = signed_hybrid, sofPower = 6, corFnc = cor, corOptions use = p and method=spearman"). The topological overall matrix was computed with the "TOMsimilarity" function (parameter TOMType = unsigned), a dynamic tree cut performed with function "cutreeDynamic" (parameter deepSplit = 2, pamRespectsDendro = FALSE), (parameter cutHeight = 0.25). Genes assigned to modules were subject to gene set overrepresentation analysis using the Metascape platform[64]. Terms with a FDR adjusted $p$-value of 0.05, a minimum

count of 3, and an enrichment factor > 1.5 (the enrichment factor is the ratio between the observed counts and the counts expected by chance) were considered.

## Associations of WGCNA module genes with IBD, CD, and UC phenotypes using MAGMA

We used MAGMA (v1.10)[32] as an alternative gene set analysis, to estimate significant associations of genes in modules with IBD, UC and CD phenotypes. For this study, we used GWAS summary statistics from de Lange et al.[65] for IBD, CD and UC phenotypes estimated for European ancestry. The GWAS variants without finished p-value were excluded. The variants were mapped to dbSNP ID using the Annovar package (avsnp147, hg38, https://annovar.openbioinformatics.org)[66]. Variants that were not mapped to dbSNP IDs were excluded. In total, 13779777 variants were used in MAGMA's gene and gene-set level analysis to estimate significant associations of GWAS variants to WGCNA modules eigengenes.

The variants were further mapped to corresponding genes in WGCNA module's gene sets using the MAGMA -annotate option and using the gene location file as an input file along with default window size. Gene based association statistics were performed on the finished $p$-value obtained from GWAS. This approach allows to combine GWAS $p$-values into a gene-level $p$-value around each gene in the specified default window size while accounting for linkage disequilibrium (LD) estimated for the European panel in the 1000 Genomes Project Phase 3[67]. While MAGMA's gene analysis is a multiple regression model, the gene-set analysis is a follow-up analysis built on the gene analysis findings for additional flexibility. Therefore, we used gene-level p-values to further estimate the significance of GWAS variants to our ME gene-set's genes. The de Lange et al.[65] meta-analysis GWAS summary statistics for IBD, CD and UC were obtained from IIBDGC genetics consortium (https://www.ibdgenetics.org/).

## CRISPR-Cas9 knockout of SBNO2 in THP-1 cells

Three guide RNA (gRNA) sequences were generated using the Integrated DNA Technologies Alt-R™ CRISPR HDR Design Tool (https://eu.idtdna.com/pages/tools/alt-r-crispr-hdr-design-tool). Oligos were purchased from Thermo Fisher and cloned into lentiCRISPR v2 as previously described[68]. LentiCRISPR v2 was a gift from Feng Zhang (Addgene plasmid # 52961; http://n2t.net/addgene:52961; RRID:Addgene_52961). Successful cloning was confirmed by PCR amplification, agarose gel electrophoresis, and Sanger sequencing. Lentiviral particles were generated and THP-1 cells transduced as described in the section "Lentiviral production and transduction". Puromycin selection was performed for 72 h, cells were kept with 0.5 μg/mL puromycin (P8833, Sigma). Following 7 days THP-1 cells were single cell cloned in 384-well plates (Corning) using irradiated (45 Gy) THP-1 cells as feeder layer. Single THP-1 clones were expanded in RPMI-1640 (Sigma) supplemented with 10% FCS (Sigma), genomic DNA extracted (omega BIO-TEK E.Z.N.A.® Tissue DNA Kit), and tested for SBNO2 knockout status by PCR, agarose gel electrophoresis, and sanger sequencing.

SBNO2 gRNA 1:
Sequence: GCGTGTCCACGATGTCCGAC
Position: 1127683
Strand: +
Oligo:
FW: caccg GCGTGTCCACGATGTCCGAC
RW: aaac GTCGGACATCGTGGACACGC c
SBNO2 gRNA 2:
Sequence: CAACGACCTCAAGTACGATG
Position: 1122526
Strand: −
Oligo:
FW: caccg CAACGACCTCAAGTACGATG
RW: aaac CATCGTACTTGAGGTCGTTG c

SBNO2 gRNA 3:
Sequence: AAAGACCTGCGACTTTGCTC
Position:
Strand: −
Oligo:
FW: caccg AAAGACCTGCGACTTTGCTC
RW: aaac GAGCAAAGTCGCAGGTCTTT c
PCR primer SBNO2 knockout validation:
gRNA1 fw: GGGTTGGTGTTTCTCACCCA
gRNA1 rw: ACTGAGACCTCACTGGACCG
gRNA2 fw: CTAGAGGCCATCACCTACGC
gRNA2 rw: TATCTGTAAGGGTGCTGGGC
gRNA3 fw: CGCCAGAATGAACCCTTAGC
gRNA3 rw: CTGAGCGAGAGAGGTCGCA

## Monocyte eQTL, isoQTL and tQTL analysis

We studied the association of the variant with alternative splicing using complementary steps including gene expression QTL (eQTL), transcript QTL (tQTL), and isoform usage QTL (isoQTL). The normalized total gene counting sequencing reads or transcripts expression values (FPKM) was regressed against genetic variants. SNPs were included in the cis analysis if they were located within 1 Mb of the gene or 100 kb of the isoform under consideration. We decomposed the gene expression matrix to the loading and score matrices. The score matrix was applied as a covariate of the Linear Model to adjust for unexplained variation in gene expression (observed dependent variable) and reveal the actual effect of genetics (categorical independent variable). Zero to 50 PCs of gene expression profiles were tested using a total of 1000 permutations[69], and the dominant PC was chosen for inclusion as covariates based on the concepts of inflection and point local maxima. False discovery rates (FDRs) were calculated using the stats R package[69]. eQTL, tQTL, and isoQTL analysis were performed with FastQTL using a linear regression[69]. Box plots and dotplots were generated using ggpubr (v0.2) and customizing ggplot2[69]. Data analysis: stats R package, Version 4.3.2; IsoformSwitchAnalyzeR, Version 2.3.0; FastQTL, Version 2.0; QTLtools, Version 1.2.

## Online resources

gnomAD: https://gnomad.broadinstitute.org/
GTEx: https://gtexportal.org/home/
GWAS Catalogue: https://www.ebi.ac.uk/gwas/
IBD Exomes Browser: https://dmz-ibd.broadinstitute.org/
LDlink: https://ldlink.nci.nih.gov
Metascape: https://metascape.org/gp/index.html
STRING−Functional Enrichment Analysis: https://string-db.org/
Open Targets Genetics: https://genetics.opentargets.org/
PhenoScanner: http://www.phenoscanner.medschl.cam.ac.uk/
SNPnexus: https://www.snp-nexus.org/v4/
hg38: https://genome.ucsc.edu/index.html

## Statistics

No statistical method was used to predetermine sample size. Statistical significance for DEGs were determined using the Wald test with all $p$-values adjusted using Benjamini-Hochberg correction. GraphPad Prism, Version 9 and 10 software, Microsoft Excel for Mac, Version 16.16.27, RStudio, Version 3.6.3, and R, Version 1.1.456 were used to assess statistical significance. All box plots depict the interquartile range as the lower and upper bounds, respectively. The whiskers represent minimum and maximum, and the centre depicts the median. All violin plots indicate quartiles and median. All statistical tests used are defined in the figure legends. $P$-values < 0.05 were considered statistically significant.

## Reporting summary

Further information on research design is available in the Nature Portfolio Reporting Summary linked to this article.

## Data availability

All sequencing data will be made freely available to organizations and researchers to conduct research following the UK Policy Framework for Health and Social Care Research via a data access agreement. Sequence data related to eQTL studies have been deposited at the European Genome–Phenome Archive, which is hosted by the European Bioinformatics Institute and the Centre for Genomic Regulation under accession no. EGAN00002778798. Raw RNA sequencing data (siRNA-mediated knockdown of SBNO2 in CD14⁺ MDM, and ectopic expression of SBNO2 isoforms in THP-1 MDM) from this study have been deposited at the European Genome–Phenome Archive, which is hosted by the European Bioinformatics Institute and the Centre for Genomic Regulation under the dataset ID EGAD50000000264. RNA sequencing data for the RISK study is deposited in the SRA database SUB6656230. Source data are provided with this paper.

## Code availability

Code related to the analyses in this study can be requested from the corresponding authors. Custom codes related to eQTL studies can be accessed at: https://bitbucket.org/bpfairfax/genetic-determinants-of-monocyte-splicing/src/master/Processing_raw_data_downstream_analyses/.

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

## Acknowledgements

We are grateful to all volunteers who contributed to this research. We like to thank Helen Su and Kazuyuki Meguro for discussions on SBNO2. We thank Melania Capitani, Aline Azabdaftari, Ying Ka Lam, Boryana Galabova and Helen Ferry for technical assistance. The Genotype-Tissue Expression (GTEx) Project was supported by the Common Fund of the Office of the Director of the National Institutes of Health, and by NCI, NHGRI, NHLBI, NIDA, NIMH, and NINDS. The data used for the analyses described in this manuscript were obtained from: the GTEx Portal on 17/09/2020. The authors would like to thank the Helmsley IBD Exomes Programme and the groups that provided exome variant data for comparison. A full list of contributing groups can be found at http://ibd.broadinstitute.org/about (IBD Exomes Portal, Cambridge, MA (URL: http://ibd.broadinstitute.org)) (accessed in July, 2021). We acknowledge the support of the Oxford Biomedical Research Centre (BRC) for the Oxford GI bio-bank (11/YH/0020, 16/YH/0247). The research was also supported by UCB Pharma. H.H.U., D.A., and I.N. are supported by the NIHR Oxford BRC, and HHU by The Leona M. and Harry B. Helmsley Charitable Trust. The views expressed are those of the author(s) and not necessarily those of the NHS, the NIHR or the Department of Health.

## Author contributions

D.A. and S.P. performed experiments. D.A., I.N. and S.V. performed bioinformatic analysis. M.P., L.D., M.A., S.K., B.F. and H.H.U. coordinated research. All authors contributed to the manuscript.

## Competing interests

H.H.U. has received research support or consultancy fees from UCB Pharma, Janssen, Eli Lilly, MiroBio, Celgene and AbbVie. D.A. was supported by a UCB Pharma fellowship. M.P., L.D., and M.A. are employed by and shareholders of UCB Pharma. S.P. is an employee and shareholder of Glaxo Smith Kline. D.A. is an employee and shareholder of Novartis Pharma AG. This article reflects the authors' personal opinions and not that of their employer. The remaining authors declare no competing interests.
