## [Peer Review File · Nature Communications]

An isoform quantitative trait locus in SBNO2 links genetic susceptibility to Crohn's disease with defective antimicrobial activityREVIEWER COMMENTS

Reviewer #1 (Remarks to the Author):

The manuscript by Aschenbrenner et al. presents an innovative approach to assigning causal mechanisms of action to genetic variants associated with human disease. By starting with genetic leads linked to Crohn's disease, the authors begin from a clinically relevant biological context and then uncover fundamental processes governing immune regulation of antibacterial responses. The authors elegantly integrate multimodal datasets to identify two distinct genetic association signals in the SBNO2 locus. The first is linked to Crohn's disease and colocalizes with epigenetic marks near the transcriptional start site of SBNO2 isoform 2. The second is linked to bone density and colocalizes with epigenetic marks near the transcriptional start site of the canonical SBNO2 isoform 1. These independent association signals were used to implicate common genetic variants associated with Crohn's disease that correlate with reduced expression (and usage) of isoform 2 in monocytes and macrophages and impaired antimicrobicidal activity. Overall, the strategy is effective and of high interest to a wide readership due to the generalizability of the approach. Moreover, this work highlights an important mechanism by which genetic variants modulate complex phenotypes through differential regulation of mRNA splicing. This is surely to translate to other variants and phenotypes. Finally, the specific finding that SBNO2 regulates innate antibacterial responses will be impactful within the immunology and infectious disease communities. The manuscript is technically sound and conclusions well supported by data.

Specific comments:

1. In general, there is an abundance of important data in extended data figures that would benefit from inclusion in the main figs. It is suggested to reconfigure and add data to main figs.
2. Addition of a separate discussion section would enhance the general impact of the conclusions.
3. Extended data Fig. 1, is a very nice presentation. It is not entirely clear what the 280 genes were or how they were selected. Review of reference 2 did not seem to clarify this point, so additional information is warranted here.
4. Extended data Fig. 7, is informative and of interest to readership due to inclusion of genes in addition to SBNO2.
5. Fig. 3a, correlation network analysis aims to identify distinct gene expression modules controlled by SBNO2 isoform 1 and 2. A more direct, empirical approach would be more compelling. This has been done by genetic knockdown or overexpression in Fig. 3c. Gene expression changes associated with these genetic perturbations are distilled into a pathway analysis in Fig. 3d-h. Some of the text description and conclusions from the pathway analyses include over-generalizations. The gene-pathway annotations are not always accurate in reference databases, and so to conclude that a pathway is active up/down based on gene expression of a subset of curated gene sets is far removed from functional evidence of pathway activity. Fig. 3h is better in that it shows which specific genes in these annotated pathways are differentially affected. Fig 4 provides more direct functional evidence that antibacterial pathways are affected by SBNO2. Suggest toning down conclusions inferred from pathway analysis and highlight more of the specific genes that are differentially regulated by SBNO2.
6. At the end of the manuscript, there is a statement about a personal communication to the effect that biallelic loss of function mutations in SBNO2 cause immunodeficiency and bone malformations. This is an important insight, but requires substantiation in the form of preprint or peer reviewed manuscript so that readers can interpret based on the totality of available evidence.

Reviewer #2 (Remarks to the Author):

In this study, Aschenbrenner et al conducted a series of comprehensive analyses investigating the role of gene SBNO2 in regulating the risk of Crohn's disease (CD). I have a few comments below that I think would significantly enhance the manuscript:

1. Initially, the authors focused on investigating the genotype-phenotype associations of the SBNO2 gene based on GWAS summary statistics, with a primary emphasis on inflammatory bowel disease (IBD), specifically CD. However, why not include ulcerative colitis (UC), another significant subtype of IBD?
2. Provide more details regarding the 176 healthy Europeans included in the QTL analyses. Specially, were they matched age, sex, and BMI (if possible)?
3. Have the authors considered performing the gene-set heritability enrichment analyses on the modules identified by WGCNA for IBD/CD/maybe UC GWAS, using e.g., stratified LDSC, MAGMA, etc, to evaluate the contribution of each module on SNP heritability for IBD/CD/UC?
4. The determination of highlighted pathways in Figure 3 is fully based on the FDR adj p-value < 0.05? Pathway analysis could lead to some results with a high false positive rate, have the authors considered including other criteria (e.g., minimum no. genes/prop. genes reported in each pathway) to make the results more robust?
5. On page 7, specifically in the conclusion paragraph, I don't agree with calling 'the two independent clusters', unless the authors provide additional details to demonstrate there are no genetic interactions between the two clusters.
6. Incorporating a flow chart at the beginning to illustrate the study's logic is advisable.

Minor:

1. Provide additional details regarding the meaning of error bars for each respective plot.
2. Figure 1a: the grey p-value scale is hard to read.
3. Figure 1c: why there is no error bar included for the bottom right plot?
4. Figure 1e (right): it is better to keep the format of the p-value the same.
5. Extended Data Figure 2: revise the font size of the figure legend.
6. Please unify the presentation of adjusted p-values in all Figures. For example, some figures (e.g., Figure 3) used 'BH' to for the adjusted p-values, while others used 'FDR'.
7. Page 3 "Comparing transcriptomic data from terminal ileum biopsy tissues from healthy individuals and patients with CD reveals ...": provide sample size for CD cases and healthy controls, are they also matched age, sex, and BMI?
8. Typo on page 10 "Only samples with a RINe \geq 9.4 were processed."
9. Typo on page 11 "Transcript to gene mapping was performed using AnnotationDbi".
10. Typo on page 12 "WGCNA analysis:".
11. WGCNA: please provide the soft power plot for selecting the soft power parameter and also the cluster dendrogram into the supplements.

We like to thank the Reviewer for positive and constructive feedback. In the revised version we have addressed all the points raised by the Reviewer.

Main changes to the manuscripts are:

1. We have restructured the figures and manuscript text.
2. We have selected key findings from the supplemental figures into the main manuscript as suggested by the Reviewer. We have included previous Extended Data Figure 7 into Figure 1c. Additionally, we have split previous Figure 3 into two figures.
3. We have newly added a MAGMA-based gene-set heritability enrichment analysis (as suggested by Reviewer 2) to Figure 3b and included previous Extended Data Figure 10e into Figure 4b.
4. We have clarified in the revised manuscript, that SBNO2 is linked to Crohn's disease but not ulcerative colitis which fits with the proposed disease mechanism of defective antimicrobial activity in macrophages.
5. We have updated Figure 1a to incorporate methylation signals described in recently published epigenetic alterations in IBD (Kalla, R. *et al.* Analysis of Systemic Epigenetic Alterations in Inflammatory Bowel Disease: Defining Geographical, Genetic and Immune-Inflammatory influences on the Circulating Methyome. *J Crohns Colitis* 17, 170-184 (2023).
6. We have revised the discussion and clarified material and methods.

REVIEWER COMMENTS

Reviewer #1 (Remarks to the Author):

The manuscript by Aschenbrenner et al. presents an innovative approach to assigning causal mechanisms of action to genetic variants associated with human disease. By starting with genetic leads linked to Crohn's disease, the authors begin from a clinically relevant biological context and then uncover fundamental processes governing immune regulation of antibacterial responses. The authors elegantly integrate multimodal datasets to identify two distinct genetic association signals in the SBNO2 locus. The first is linked to Crohn's disease and colocalizes with epigenetic marks near the transcriptional start site of SBNO2 isoform 2. The second is linked to bone density and colocalizes with epigenetic marks near the transcriptional start site of the canonical SBNO2 isoform 1. These independent association signals were used to implicate common genetic variants associated with Crohn's disease that correlate with reduced expression (and usage) of isoform 2 in monocytes and macrophages and impaired antimicrobicidal activity. Overall, the strategy is effective and of high interest to a wide readership due to the

generalizability of the approach. Moreover, this work highlights an important mechanism by which genetic variants modulate complex phenotypes through differential regulation of mRNA splicing. This is surely to translate to other variants and phenotypes. Finally, the specific finding that SBNO2 regulates innate antibacterial responses will be impactful within the immunology and infectious disease communities. The manuscript is technically sound and conclusions well supported by data.

Specific comments:

1. In general, there is an abundance of important data in extended data figures that would benefit from inclusion in the main figs. It is suggested to reconfigure and add data to main figs.

We thank the Reviewer for this suggestion. We have restructured the figures and manuscript text. We have included previous Extended Data Figure 7 into Figure 1c. Additionally, we have split previous Figure 3 into two figures. We have newly added a MAGMA-based gene-set heritability enrichment analysis (as suggested by Reviewer 2) to Figure 3b and included previous Extended Data Figure 10e into Figure 4b.

2. Addition of a separate discussion section would enhance the general impact of the conclusions.

Thank you for this suggestion. We have now added a separate discussion section to the updated manuscript version.

The section reads:

To unravel the mechanistic complexity of multigenic disorders such as IBD⁴⁵ represents a prerequisite for personalized medicine where those patients that are most likely to respond receive disease- and context-specific treatment. Patient care in rare monogenic diseases aims for precision medicine and studies focusing on the elucidation of the pathogenesis and mechanistic basis continuously inform key checkpoints of the immune system and enhance our understanding of health and disease, also in IBD²². Additionally, genetically informed models provide potential for accurate definition of novel drug targets and identification of key pathways for successful therapeutic interventions. Still, the mechanistic basis for genetic associations in immune mediated diseases, including classical IBD, remains to be fully elucidated. Furthermore, previous studies have preferentially focused on understanding genetic disease risk at the canonical gene level and demonstrated the link between GWAS signals and eQTLs, and the importance of cell-type or cell-state specific regulation, but largely ignored the complexity of transcript level regulation. The distinct role of sQTLs in the genetic regulation of transcription and complex trait variation has only been appreciated more recently^{46, 47, 48, 49, 50}.

In this study we investigated the genetic and transcriptional regulation of the GWAS-identified IBD risk gene *SBNO2* and the related gene network in human monocytes and macrophages. Our analysis revealed differential expression of *SBNO2* isoforms in dependence of exogenous IL-10 stimulation and also LPS-induced autocrine IL-10 signalling that specifically increased *SBNO2* ISO2 expression. We showed that 15 high confidence IBD risk genes changed transcript usage or expression in LPS or LPS + aIL-10R stimulated MDM, while *SBNO2* represented the one high confidence IBD risk gene expressed in MDM that changed transcript usage and transcript expression upon IL-10 stimulation. In addition, we demonstrated the induction of *SBNO2* ISO2 expression downstream of IL-10 but not IL-6 signalling, suggestive of the presence of a yet unidentified mechanism of selectivity that discriminates anti-inflammatory (IL-10-dependent) from pro-inflammatory (IL-6-dependent) STAT3 signalling at the target gene locus, adding an additional layer of control to the regulation of *SBNO2* gene expression.

Based on the analysis of genetic regulatory effects in 176 healthy donors steady state and LPS-stimulated monocytes we showed reduced *SBNO2* expression, *SBNO2* ISO2 expression, and *SBNO2* ISO2 transcript usage upon LPS stimulation, providing a mechanistic basis for the association of genetic variation at the *SBNO2* locus with Crohn's disease. Furthermore, combination of WGCNA³² and MAGAMA³³ analyses highlighted the specific association of an IL-10-driven *SBNO2* ISO2-specific gene network with CD heritability that was functionally associated with pathways relevant for bacterial handling. Although we initially hypothesized that LOF in *SBNO2* would lead to de-regulated inflammatory responses by disrupting IL-10-dependent suppressive mechanisms we could not confirm this assumption. Instead functional characterization of MDM bacterial handling capacity in gentamycin protection assays using ectopic expression and CRISPR-Cas9 knockout in THP-1 MDM point towards a causal role of immunodeficiency due to decreased anti-microbial activity in the development of intestinal inflammation in patients with GWAS-associated genetic variants that are linked to decreased expression or usage of *SBNO2* ISO2. On this basis it is tempting to speculate that epigenetic regulation and disease-associated genetic variation at the *SBNO2* locus may impact on the prevalence of *SBNO2* isoform transcription, and ultimately impact on cellular function. Furthermore, these results may also explain the immunodeficiency and infection susceptibility observed in patients with LOF in *SBNO2*⁴⁴.

In contrast to *SBNO2* ISO2, *SBNO2* ISO1 expression did not depend on IL-10 signalling but on LPS stimulation. Variants located in proximity to the *SBNO2* ISO1 TSS associated with changes in HBMD, which may relate to *SBNO2*'s role in regulating osteoblast and osteoclast function. Indeed, increased expression of *SBNO2* has been found in male ankylosing spondylitis associated osteoporosis⁵¹. Previous studies performed in *Sbno2* knockout mice have revealed a regulatory function in osteoclast differentiation¹⁶ and knockdown and ectopic expression experiments presented in this study support a role

for SBNO2 in the regulation of human osteoclast multinucleation and differentiation and support evolutionary conserved functions of SBNO2 in normal bone development. One of the top SBNO2 positively regulated genes across stimulation conditions was *OCSTAMP*, a gene coding for a surface receptor involved in osteoclast differentiation with biologic functions similar to DC-STAMP^{52, 53}, that was found regulated by SBNO2 in the murine gene knockout model¹⁶. The involvement of SBNO2 in the positive and negative regulation of osteoclast and osteoblast function respectively suggests a role in maintaining the optimal balance of osteoclast *versus* osteoblast differentiation transcriptional programs that orchestrate bone reduction *versus* formation and may explain the presence of retained teeth in patients with rare disease caused by LOF in SBNO2⁴⁴.

While we have focused our analysis on monocytes and macrophages, this study also highlights potential roles of SBNO2 in the differentiation and function of other cell types, in particular neutrophils. Indeed, the impact of SBNO2 on the expression of CXCR2 and CXCR4, and genes involved in the oxidative stress response (e.g. *SOD2*, *GPX3*) suggests an additional role in the regulation of biologic processes that are key to normal neutrophil function. Normal CXCR2 and CXCR4 expression in neutrophils is required for the optimal maturation and release of neutrophils from bone marrow into circulation⁵⁴. An impairment of CXCR2 signalling in neutrophils would likely cause the premature release of dysfunctional neutrophils into circulation⁵⁴, a testable hypothesis that requires confirmation in the context of rare human disease. In addition, impaired oxidative stress response in neutrophils would lead to reduced elimination of radical oxygen species thereby allowing increased damage to host tissue during an immune response. Furthermore, the decreased production of hypochlorous acid from superoxide would additionally reduce the antibacterial activity of neutrophils⁵⁵.

In conclusion, we demonstrate that SBNO2 regulates key cellular properties of human myeloid cells in inflammation, anti-microbial response, and osteoclast differentiation and provide a mechanistic basis for the identified association of human genetic variation at the *SBNO2* locus with increased risk for the development of chronic intestinal inflammation and reduced bone density.

3. Extended data Fig. 1, is a very nice presentation. It is not entirely clear what the 280 genes were or how they were selected. Review of reference 2 did not seem to clarify this point, so additional information is warranted here.

We have added a more detailed description on the source of the list of high confidence IBD loci published in reference 2 (Bolton et al. An Integrated Taxonomy for Monogenic Inflammatory Bowel Disease. *Gastroenterology*. 2022 Mar;162(3):859-876. doi: 10.1053/j.gastro.2021.11.014. Epub 2021 Nov 13.

Erratum in: *Gastroenterology*. 2022 Jun;162(7):2143. PMID: 34780721) in the legend to Extended Data Figure 1.

Page 31:

Extended Data Figure 1: GTEx eQTL, sQTL, ieQTL, and isQTL analysis of confident GWAS IBD genes.

(a) Heatmap presentation of the presence or absence of GTEx eQTL, sQTL, ieQTL, and isoQTL across confident GWAS IBD genes² (n = 280, including SBNO2, Bolton C. et al., *Gastroenterology* 2022, Supplementary Table 4: Genes included in the analysis of the intersection of monogenic IBD genes and polygenic IBD loci). (b) Pie chart presentation of numbers of confident GWAS IBD genes according to (a).

4. Extended data Fig. 7, is informative and of interest to readership due to inclusion of genes in addition to SBNO2.

We thank the Reviewer for this comment and we have now moved Extended Data Figure 7 into the main Figure 1c.

Figure 1

5. Fig. 3a, correlation network analysis aims to identify distinct gene expression modules controlled by SBNO2 isoform 1 and 2. A more direct, empirical approach would be more compelling. This has been done by genetic knockdown or overexpression in Fig. 3c. Gene expression changes associated with these genetic perturbations are distilled into a pathway analysis in Fig. 3d-h. Some of the text description and conclusions from the pathway analyses include over-generalizations. The gene-pathway annotations are not always accurate in reference databases, and so to conclude that a pathway is active up/down based on gene expression of a subset of curated gene sets is far removed from functional evidence of pathway activity. Fig. 3h is better in that it shows which specific genes in these annotated pathways are differentially affected. Fig 4 provides more direct functional evidence that antibacterial pathways are affected by SBNO2. Suggest toning down conclusions inferred from pathway analysis and highlight more of the specific genes that are differentially regulated by SBNO2.

We thank the reviewer for this suggestions. We have rephrased the respective paragraphs to put more emphasis on individual genes rather than pathway terms.

Page 6 and 7:

Next, we assigned biologic functions to those modules that were correlated with *SBNO2* ISO1 expression only (ME9, ME15), *SBNO2* ISO2 expression only (ME12, ME13), and those modules that were correlated with *SBNO2* gene expression (ME6, ME14) (**Figure 3c, Extended Data Figure 9**). Consistent with the identity of WGCNA modules ME9 and ME15 highly correlated (module correlation > 0.5) member genes (e.g. *HLA*, *IFITM2*, *IFIT1*, *IFIT3*, *IFNAR2*, *IFNGR2*, *DAP*, *STAT1*, *STAT2*, *STAT5A*, *CCL3*, *CCL5*, *CCL18*, *NFKB1*, *NFKB2*, *MYD88*, *IRF1*, *IRF2*, *IRF7*, *IRF9*, *OAS1*, *OAS2*, *OAS3*, *IFIH1*, *IFI35*, *RELB*; **Extended Data Table 3**), *SBNO2* ISO1-expression-correlated modules were linked to pathways of inflammatory responses (e.g. cytokine and interferon signalling). *SBNO2* ISO2 expression, that was correlated with ME12 and ME13 member genes (e.g. *BCL2L1*, *CTSL*, *PIK3CD*, *PTEN*, *BNIP3L*, *CAPN1*, *CAPNS1*, *GAPDH*, *GNAI3*, *HSPB1*, *SREBF1*, *GPR137B*, *ATP13A2*, *RAB3GAP2*, *SNX6*, *LGALS8*, *RAB7A*, *AP3D1*, *CSNK1D*) associated with pathways linked to antimicrobial activity (e.g. autophagy, lysosome) (**Figure 3c and Extended Data Table 3**). *SBNO2* gene expression was correlated with mRNA metabolic processes, JAK-STAT signalling, and GTPase signalling (**Figure 3c**).

To understand SBNO2's impact on MDM function in more detail we performed siRNA-mediated knockdown and RNA-seq (**Figure 4a and b, Extended Data Figure 10a-d**). Interestingly, we found known IL-10-signalling suppressed genes downregulated upon knockdown of *SBNO2* (e.g. *IL23A*, *IL20*, and *IL24*³⁴, **Figure 4a and b**). Furthermore, the expression of chemokine receptors *CXCR2* and *CXCR4*, that influence granulocyte and myeloid tissue homing and function^{35, 36} were negatively regulated upon

SBNO2 knockdown. Several genes involved in the regulation of osteoblast and osteoclast function, including *KREMEN1*, a regulator of bone formation^{37, 38}, that was found upregulated upon *SBNO2* knockdown (**Figure 4b**), were found deregulated upon *SBNO2* knockdown (Upregulated: *CSF1R*, *CYP27B1*, *FAM20C*, *GREM1*, *LMNA*, *LRP5*, *SORT1*, *SPP1*, *TNEN119*, and *TNS3*; Downregulated: *THBS1*, *ACP5*, *BMP6*, *CA2*, *DUSP4*, *INHBA*, *NFIL3*, and *VDR*; **Extended Data Table 5**). These results indicated that *SBNO2* may not mediate the conventional anti-inflammatory effects of IL-10.

To identify perturbed functional pathways, we performed an enrichment analysis across stimulation conditions and those genes that were found differentially regulated by *SBNO2* knockdown (**Figure 4c-g**, **Extended Data Table 6-9**). We identified condition-specific and shared pathways upon LPS stimulation in the presence or absence of *SBNO2* ISO2-inducing IL-10 signalling (**Figure 4c**). In LPS- and LPS+aIL-10R-stimulated MDM *SBNO2* knockdown-downregulated genes (*SBNO2*-induced genes) contained *THBS1* (Thrombospondin 1, TSP1) as top regulated gene (**Figure 4d and 4g**). Functionally, TSP1 has been linked to bone development, normal lung homeostasis³⁹, and synaptogenesis in astrocytes⁴⁰. At the pathway-level *SBNO2* knockdown-downregulated genes in LPS- and LPS+aIL-10R-stimulated MDM associated with the term “p53 signalling pathway”. Strikingly, in the context of LPS but not LPS+aIL-10R stimulation *SBNO2*-induced genes (e.g. *ATP6V1A*, *ATP6V1B2*, *ATP6V1D*, *ATP6V1F*, *ATP6V1G1*, *ATP6V1H*, *ATP6V1E1* and 2, *ATP6V0A1* and 2, *ATP6V0B*, *ATP6V0C*, *ATP6V0D1* and 2, *ATP6V0E1* and 2, *ATP6AP1*, *TCIRG1*, *MCOLN1*, *LMTK2*, *ARHGAP1*, *RAB11B*, *TFRC*, *DNM2*, *STEAP3*, *ATP6V1C1*, *TF*, *HFE*, *CLTC*) associated with the term “Transferrin transport” as the top regulated pathway (**Figure 4d and 4g**), suggesting that IL-10-induced *SBNO2* expression promoted ATPase function, that is a critical mechanisms in endocytosis, lysosomal function, phagosome acidification, and bone resorption⁴¹. Interestingly, *SBNO2* has previously been identified as a regulator of autophagy-dependent intracellular pathogen defence in a GWAS-based IBD genes siRNA screen⁴². In contrast, *SBNO2*-suppressed genes and pathways largely overlapped between LPS and LPS+aIL-10R conditions (**Figure 4e**) and included the regulation of pattern recognition receptor (PRR) signalling (**Figure 4e**) by direct negative regulation of PRRs *TLR1*, *TLR2*, *TLR4*, *TLR6* and *NOD2* expression (**Figure 4g**). LPS+aIL-10R stimulation-regulated genes pathway term enrichment that was not shared with LPS stimulation included terms linked to complement activation and included genes such as *C2*, *C3*, *C4A*, *CFB* and *CFP* (**Figure 4e and 4g**). In addition, *SBNO2* knockdown deregulated LPS-specific, LPS+aIL-10R-specific, and shared genes associated with functional pathways of the acute phase response (*SBNO2*-induced: *STAT3*, *HAMP*, *SAA1*, *SAA2*, *LBP*; *SBNO2*-suppressed: *IL1A*, *FNI*, *IL1B*, *ITIH4*, *IL6*), signal transducer and activator of transcription (STAT)-dependent cytokine signalling (*SBNO2*-induced: *PECAM1*, *LYN*, *IL15*, *JAK2*, *CSFR1*; *SBNO2*-suppressed: *IL20*, *CSF2*, *IL23A*, *IL24*, *LIF*), and chemokine secretion and sensing

respectively (SBNO2-induced: *CXCL9*, *CXCL13*, *CCL1*; SBNO2-suppressed: *CXCR2*, *CXCR4*, *CCL22*) (Figure 4f and 4g).

6. At the end of the manuscript, there is a statement about a personal communication to the effect that biallelic loss of function mutations in SBNO2 cause immunodeficiency and bone malformations. This is an important insight, but requires substantiation in the form of preprint or peer reviewed manuscript so that readers can interpret based on the totality of available evidence.

We thank the Reviewer for highlighting this point. We have now added a reference to the conference abstract where these data were presented in an oral presentation (available as video presentation and abstract).

Jing H, Dove C, Zhang Y, Price S, Koneti R, Su HC, editors. Novel immunodysregulation disorder caused by loss-of-function mutations in SBNO2. Clinical Immunology Society 2016 Annual Meeting; 2016; Boston, Massachusetts, USA.

4473: NOVEL IMMUNODYSREGULATION DISORDER CAUSED BY LOSS-OF-FUNCTION MUTATIONS IN SBNO2

Huie Jing¹, Christopher Dove, BS¹, Yu Zhang, PhD¹, Susan Price², Rao Koneti, MD² and Helen C. Su¹
¹Laboratory of Host Defenses, National Institute of Allergy and Infectious Diseases, National Institutes of Health, Bethesda, MD, ²Laboratory of Clinical Infectious Diseases, National Institute of Allergy and Infectious Diseases, NIH, Bethesda, MD

We studied an 8-year-old girl who had a history of transfusion-dependent anemia, thrombocytopenia, leukocytosis, splenomegaly, severe pneumonias, and multiple episodes of respiratory failure. Vaccine titers waned over time and hypogammaglobulinemia developed. To identify a genetic cause responsible for immune dysregulation, we performed comparative genomic hybridization array, which revealed a 120 kB heterozygous deletion on chromosome 19. Targeted gene sequencing of other alleles revealed a splicing mutation in a gene encoding strawberry notch homologue 2 (SBNO2), which belongs to a novel member of the Notch family of transcriptional repressors. The loss-of-function mutation in the patient was confirmed by western blotting. Overexpression of this gene showed nuclear translocation, suggesting a function in transcription regulation. An anti-inflammatory effect was previously reported through repression of inflammatory gene transcription. However, we did not observe any difference in transcriptional profiles between monocytes from the patient as compared to normal controls. qRT-PCR showed that SBNO2 was highly expressed in myeloid lineages. Ongoing work will address if SBNO2 affects gene expression in neutrophils. Identification of potential gene targets will help us understand the

function of this gene and potentially explain the disease pathogenesis in the patient.

<https://link.springer.com/article/10.1007/s10875-016-0237-x>

The authors have recently presented an updated work Kazuyuki, M., et al. (2022). Unwinding the molecular pathogenesis of a novel inherited immunodysregulatory disorder caused by loss-of-function mutations in DExD/H box helicase SBNO2. The 9th Global Network Forum on Infection and Immunity, Japan. We have also included a respective reference.

https://jglobal.jst.go.jp/en/detail?JGLOBAL_ID=202302213421280661

A preprint or manuscript has not yet been published.

Reviewer #2 (Remarks to the Author):

In this study, Aschenbrenner et al conducted a series of comprehensive analyses investigating the role of gene SBNO2 in regulating the risk of Crohn's disease (CD). I have a few comments below that I think would significantly enhance the manuscript:

1. Initially, the authors focused on investigating the genotype-phenotype associations of the SBNO2 gene based on GWAS summary statistics, with a primary emphasis on inflammatory bowel disease (IBD), specifically CD. However, why not include ulcerative colitis (UC), another significant subtype of IBD?

We thank the Reviewer for highlighting this important point. We have focused on IBD and specifically CD in our analysis, since genetic variation in SBNO2 is linked to IBD and CD but not UC. To better clarify this point we have added the following statement to the text:

Page 2:

The genetic locus that covers the Strawberry notch homologue 2 (*SBNO2*) gene, represents a high confidence association for IBD, specifically Crohn's disease (CD)^{3,4,5} but not ulcerative colitis (UC).

2. Provide more details regarding the 176 healthy Europeans included in the QTL analyses. Specially, were they matched age, sex, and BMI (if possible)?

In Extended Data Table 2 (EDT2_QTL analyses donor information) we do now provide age, sex, and BMI parameters for the 176 healthy donors who were recruited for this study via the Oxford biobank.

3. Have the authors considered performing the gene-set heritability enrichment analyses on the modules identified by WGCNA for IBD/CD/maybe UC GWAS, using e.g., stratified LDSC, MAGMA, etc, to evaluate the contribution of each module on SNP heritability for IBD/CD/UC?

We thank the Reviewer for this suggestion. We have now performed the gene-set heritability enrichment analysis and added the MAGMA-based results to Figure 3b and Extended Data Table 4.

Page 5 and 6:

MAGMA-based³³ gene-set heritability analysis revealed significant associations of genes in modules ME1, ME5, ME9, and ME15 with IBD and UC, while ME1, ME12, ME13, ME15, and ME17 showed significant associations with CD suggesting that these modules are relevant for disease development (**Figure 3b, Extended Data Table 4**). Interestingly, WGCNA modules ME12 and ME13 that were correlated with IL-10 stimulation in MDM, and that were associated with *SBNO2* ISO2 expression but

not *SBNO2* gene or *SBNO2* ISO1 expression, were significantly associated with CD but not IBD or UC. (**Figure 3a**). In agreement with the CD-specific association of *SBNO2* genetic risk these results show that IL-10 driven *SBNO2* ISO2 expression in MDM is linked to gene expression programs that specifically associate with CD development.

Figure 3

4. The determination of highlighted pathways in Figure 3 is fully based on the FDR adj p-value < 0.05? Pathway analysis could lead to some results with a high false positive rate, have the authors considered including other criteria (e.g., minimum no. genes/prop. genes reported in each pathway) to make the results more robust?

We thank the reviewer for highlighting this important point. Pathway enrichments shown in Figure 3b and Extended Data Figure 9 were performed using the Metascape platform (Zhou Y, Zhou B, Pache L, Chang M, Khodabakhshi AH, Tanaseichuk O, Benner C, Chanda SK. Metascape provides a biologist-oriented resource for the analysis of systems-level datasets. *Nat Commun.* 2019 Apr 3;10(1):1523. doi: 10.1038/s41467-019-09234-6. PMID: 30944313; PMCID: PMC6447622.). Terms with a FDR adjusted p-value < 0.05, a minimum count of 3, and an enrichment factor > 1.5 (the enrichment factor is the ratio between the observed counts and the counts expected by chance) were considered. We have added a statement to the Materials and Methods section.

Page 15:

Genes assigned to modules were subject to gene set overrepresentation analysis using the Metascape platform⁵¹. Terms with a FDR adjusted p-value < 0.05, a minimum count of 3, and an enrichment factor > 1.5 (the enrichment factor is the ratio between the observed counts and the counts expected by chance) were considered.

5. On page 7, specifically in the conclusion paragraph, I don't agree with calling 'the two independent clusters', unless the authors provide additional details to demonstrate there are no genetic interactions between the two clusters.

We thank the reviewer for providing feedback on this point. We do agree that the term independent is not accurate in this context and therefore adapted the statement.

Page 3:

CD only associated variants and those variants associated with CD and HBMD formed two clusters of pairwise linkage disequilibrium (Extended Data Figure 2) across human populations indicating differential inheritance and selection.

6. Incorporating a flow chart at the beginning to illustrate the study's logic is advisable.

Thank you for this suggestion. We have now generated a flow chart to illustrate the study logic and replaced the previous graphical abstract. We have integrated the previous graphical abstract into Figure 5.

Page 57 and page 34:

Graphical Abstract

Figure 5

Minor:

1. Provide additional details regarding the meaning of error bars for each respective plot.

We thank the Reviewer for raising this point. We have now added the respective information to relevant Figure legends.

2. Figure 1a: the grey p-value scale is hard to read.

We thank the Reviewer for highlighting this point. We have adjusted the grey color scale to black.

3. Figure 1c: why there is no error bar included for the bottom right plot?

In Figure 1c bottom right the error bars are smaller than the individual symbols and are therefore overlapping with the symbols in front.

4. Figure 1e (right): it is better to keep the format of the p-value the same.

We thank the Reviewer for pointing this out. We have adjusted the display of the p-value for Figure 1e on the right.

5. Extended Data Figure 2: revise the font size of the figure legend.

Thank you. We have corrected this.

6. Please unify the presentation of adjusted p-values in all Figures. For example, some figures (e.g., Figure 3) used 'BH' to for the adjusted p-values, while others used 'FDR'.

Thank you for highlighting this inconsistency. We have corrected this in the updated manuscript version.

7. Page 3 “Comparing transcriptomic data from terminal ileum biopsy tissues from healthy individuals and patients with CD reveals ...”: provide sample size for CD cases and healthy controls, are they also matched age, sex, and BMI?

Thank you for this comment. In Extended Data Table 1 (EDT1_RISK cohort patient information) we do now provide this information.

8. Typo on page 10 “Only samples with a RINe \geq 9.4 were processed.”.

Thank you. We have corrected this.

9. Typo on page 11 “Transcript to gene mapping was performed using AnnotationDbi”.

Thank you. We have corrected this.

10. Typo on page 12 “WGCNA analysis:”.

Thank you. We have corrected this.

11. WGCNA: please provide the soft power plot for selecting the soft power parameter and also the cluster dendrogram into the supplements.

We have now added an additional Extended Data Figure 8 presenting the WGCNA soft power plot, the cluster dendrogram, and the eigengene dendrogram.

Extended Data Figure 8

a

b

c

REVIEWERS' COMMENTS

Reviewer #1 (Remarks to the Author):

The authors have satisfactorily addressed all of my comments. I commend them on their nice work.

Reviewer #2 (Remarks to the Author):

I am grateful to the authors for taking the time to address each of my comments. The authors have solved most of my comments. I only have one quick question about Figure 3b: I understand that "Betas" and "-log10 p-values" correspond to the MAGMA results. But what does the additional "P-value" mean (marked with a red circle < 0.05)?

We like to thank the Reviewer again for positive and constructive feedback. In the revised version we have addressed the point raised by the Reviewer.

Main changes to the manuscripts are:

1. Adjustments to Figure 3b and the respective figure legend.

Additional changes:

2. Added Source Data files:

Source Data Figure 1b, Source Data Figure 1c, Source Data Figure 1d, Source Data Figure 1e, Source Data Figure 1f, Source Data Figure 2b, Source Data Figure 3b, Source Data Figure 3c, Source Data Figure 4b, Source Data Figure 5a, Source Data Figure 5b, Source Data Figure 5c, Source Data Figure 5d, Source Data Figure 5e, Source Data Figure 5f, Source Data Supplementary Figure 1a, Source Data Supplementary Figure 4a, Source Data Supplementary Figure 4b, Source Data Supplementary Figure 4c, Source Data Supplementary Figure 5a, Source Data Supplementary Figure 5b, Source Data Supplementary Figure 7a, Source Data Supplementary Figure 7b, Source Data Supplementary Figure 9, Source Data Supplementary Figure 10a, Source Data Supplementary Figure 10b, Source Data Supplementary Figure 11a, Source Data Supplementary Figure 11b, Source Data Supplementary Figure 11c, Source Data Supplementary Figure 11d, Source Data Supplementary Figure 11e.

3. Added missing statistics description in figure legends: Figure 2b, Figure 1d, Figure 5a, Figure 5c, Figure 5e.
4. Updated Figure 5a and the legend to Figure 5a.
5. Added exact values for p-values across all figures.

REVIEWER COMMENTS

Reviewer #1 (Remarks to the Author):

The authors have satisfactorily addressed all of my comments. I commend them on their nice work.

Reviewer #2 (Remarks to the Author):

I am grateful to the authors for taking the time to address each of my comments. The authors have solved most of my comments. I only have one quick question about Figure 3b: I understand that “Betas” and “-log₁₀ p-values” correspond to the MAGMA results. But what does the additional “P-value” mean (marked with a red circle < 0.05)?

We thank the Reviewer for highlighting this point that may cause misinterpretation of Figure 3b. Red borders around individual dots indicate enrichment p-values < 0.05 . These are transformed as $-\log_{10}$ p-values for dot size presentation.

To avoid confusions, we have removed the additional label for red borders from the figure and instead describe these in the figure legend:

Figure 3: Definition of functional gene modules associated with SBNO2 gene and isoform expression in human primary MDM. (a) Correlation of the 17 identified WGCNA modules of genes (y-axis) with stimulations, genes, and transcripts of interest (x-axis). Correlation coefficients and p-values are indicated for modules with $p\text{-values} \leq 0.05$ (Pearson correlation, two-sided test, uncorrected p-values). (b) MAGMA-based³³ gene-set heritability analysis on WGCNA modules for IBD, CD, and UC⁶⁷. The dot blot illustrates the directionality of association as color code and the enrichment significance ($-\log_{10}$ pvalue) as dot size. Red circular borders around individual dots indicate $p\text{-values} < 0.05$ (one-sided test, uncorrected p-values, as implemented in MAGMA³³). (c) Enrichment of GO pathway terms based on those gene modules that were found correlated with ISO1 (ME9 and ME15), ISO2 (ME12 and ME13, and ISO1 and ISO2 (ME6 and ME14) expression (hypergeometric test, Bonferroni correction, $p_{adj} < 0.05$). Source data are provided as a Source Data file for Figures 3b and 3c.

Figure 3